

# A Scalability Study of the Ice-sheet and Sea-level System Model (ISSM, Version 4.18)

Yannic Fischler[1], Martin Rückamp[2], Christian Bischof[1], Vadym Aizinger[3], Mathieu Morlighem[4,5], and Angelika Humbert[2,6]

[1]Department of Computer Science, Technical University Darmstadt, Darmstadt, Hesse, Germany
[2]Alfred-Wegener-Institut, Helmholtz-Zentrum für Polar- und Meeresforschung, Bremerhaven, Bremen, Germany
[3]Chair of Scientific Computing, University of Bayreuth, Bayreuth, Bavaria, Germany
[4]Department of Earth Sciences, Dartmouth College, Hanover, United States of America
[5]Department of Earth System Science, University of California Irvine, United States of America
[6]Faculty of Geosciences, University of Bremen, Bremen, Germany

**Correspondence:** Yannic Fischler (yannic.fischler@tu-darmstadt.de)

**Abstract.** Accurately modeling the contribution of Greenland and Antarctica to sea level rise requires to solve partial differential equations at a high spatial resolution. It is important to test the scalability of existing ice sheet models in order to assess whether they are ready to take advantage of new cluster architectures. In this paper, we discuss the overall scaling of the Ice-sheet and Sea-level System Model (ISSM) applied to the Greenland ice sheet. The model setup used as benchmark problem comprises a variety of modules with different levels of complexity and computational demands. The core builds the so-called stress balance module, which uses the higher-order approximation (or Blatter-Pattyn) of the Stokes equations and a mesh of linear prismatic finite elements to compute the ice flow. We develop a detailed user-oriented, yet low-overhead performance instrumentation tailored to the requirements of earth system models and run scaling tests up to 6 144 MPI processes. The results show that the computation of the Greenland model scales overall well up to 3 072 MPI processes, but is eventually slowed down by matrix assembly, the output handling, and lower-dimensional problems that employ lower numbers of unknowns per MPI process. We also discuss improvements of the scaling and identify further improvements needed for climate research. The instrumented version of ISSM, thus, not only identifies potential performance bottlenecks that were not present at lower core counts but also provides the capability to continually monitor the performance of ISSM code basis. This is of long-term significance as the overall performance of ISSM model depends on the subtle interplay between algorithms, their implementation, underlying libraries, compilers, run-time systems and hardware characteristics, all of which are in a constant state of flux.

## 1 Introduction

Projections of future sea level rise are a major societal demand. The future mass loss of ice sheets and glaciers is one of the primary sources of sea level rise (Church et al., 2013). Today, projections are still subject to large uncertainties, which stem in



particular from the climate forcing and ice-sheet model characteristics (such as the initial state, Goelzer et al., 2020). While the

momentum balance can nowadays be solved using a higher-order approximation (HO, also called Blatter-Pattyn, Blatter, 1995;

Pattyn, 2003) representing the physical system reasonably well, the benefits of HO pay out only if the resolution is sufficiently

high, especially in the vicinity of the grounding line where the ice sheet goes afloat. The grid resolution requirement comes

with additional computational costs, a limiting factor that needs to be overcome. In addition, standalone ice sheet projections

are suffering from a large spread in climate forcing fields (e.g. Goelzer et al., 2020). As a consequence, atmosphere and ocean

models will be run in the future at higher spatial resolutions, so when ice sheet codes are coupled with other components

of Earth System Models (ESMs), they need to reach a level of computational performance – especially in terms of parallel

scalability – comparable to that of ocean and atmosphere models.

Since complex bed topographies, rugged coastlines of ice sheets and small scale features form an irregular geometry (e.g.

narrow confined fjords in Greenland or small pinning points in Antarctic ice shelves), unstructured meshes are best suited.

This motivated the development of codes based on finite element and finite volume discretizations with triangular or Voronoi

meshes (Larour et al., 2012; Gagliardini et al., 2013; Hoffman et al., 2018b; Berends et al., 2020) – such as the Ice-sheet and

Sea-level System Model (ISSM, Larour et al., 2012). The flexible multi-physics model ISSM provides full Stokes, HO, shallow

shelf, and shallow ice approximations for the momentum balance making it particularly versatile and flexible.

To study the performance of ISSM, we select a real-life system as a test case: the Greenland Ice Sheet (GrIS) simulated

at different horizontal resolutions – covering the range from what is today's standard for long-term simulations (Plach et al.,

2019), such as paleo-spinups, up to the present-day highest resolutions in projections such as the ones used in the international

benchmark experiment ISMIP6 (Goelzer et al., 2020; Rückamp et al., 2020a). This way we contribute a 'user-oriented' perfor-

mance analysis tailored to the requirements of ESMs. Such an in-depth performance analysis has not yet been performed for

ISSM and comes timely with the first exascale-ready ESMs (Golaz et al., 2019) ushering a new era of climate modelling.

Several metrics have been proposed to quantify 'scalability', that is the performance response of a code when additional

hardware resources are made available (see, e.g. Perlin et al. (2016)). 'Strong scaling' shows the performance of a fixed-size

problem when additional cores are employed to solve it. The baseline measurement is, in theory (as in Ahmdahl's law), the

run-time with one core, but this is infeasible for many realistic model setups due to memory and run-time limitations. 'Weak

scaling' (or Gustavson's law), on the other hand, reflects the fact that usually the problem size increases with the size of the

compute system. In our study, we mainly investigate how ISSM performs on a particular, fixed-size model setup using from 96

to 6144 cores, i.e. over almost two orders of magnitude in core count, but we also show performance for various resolutions of

the same domain. These results give us insight into both strong and weak scaling issues and allow us, in particular, to identify

the simulated years per (wall-clock) day (SYPD), which is an important measure for ESMs.

Numerical models such as the ISSM are generally based on a discretization of the underlying system of partial differential

equations (PDEs) leading to a fully discrete system of nonlinear or linear algebraic equations to be solved. In the context of

finite elements, the computational domain is partitioned using a computational mesh consisting of elements (e.g. triangles or

quadrilaterals in the 2D, tetrahedra or prisms in the 3D), and the discrete unknowns are specified per node, per element or per

face of the mesh, depending on the specific finite element scheme and used approximation order. The discrete unknowns are





called degrees of freedom (DOF), and their total number then specifies the size of the discrete linear or nonlinear system to be solved. With increasing resolution as well as increasing order of shape functions, the number of DOF is increasing for a particular PDE boundary value problem. In parallel computing, the problem is distributed over a number of cores, leaving each core with a particular number of DOF/core. Studies on performance in other disciplines have shown (Bn'a et al., 2020) that a certain minimum number of DOF/core is required to ensure good performance. For the software package PETSc (Balay et al.,

2021a) on which ISSM is built, the recommendation is a minimum of 10 000 DOF/core.

The employed model setup in our study comprises a variety of modules with different levels of complexity. Sophisticated performance analyses that identify the impact of such a multi-physics problem for overall code performance do exist for ocean models (Reuter et al., 2015; Huang et al., 2016; Prims et al., 2018; Koldunov et al., 2019) and, in somewhat less detail, for the atmosphere (Neumann et al., 2019); however, they are not yet standard for ice sheet codes. Most studies analyse

the performance of the stationary Stokes problem. For MALI[1] (MPAS-Albany Land Ice, HO, finite elements, Hoffman et al., 2018a) this was presented by (Tezaur et al., 2015a, b) and revealed a good overall scalability. Similarly, Gagliardini et al. (2013) achieved a good weak and strong scaling efficiency for Elmer/Ice (full Stokes finite elements). In a transient setup, Dickens (2015) analysed the scalability and performance of PISM's (Parallel Ice Sheet Model, Hybrid SIA+SSA, finite differences, Bueler and Brown, 2009) compute and output phases (i.e. I/O) in a transient setup. They found the I/O component to be a

considerable bottleneck while the compute phase scaled well. As we will see, however, for sophisticated multi-physics and multi-resolution codes such as ISSM, a finer grained analysis is necessary to shed light on the drivers of scalability issues (see also (Chang et al., 2018)).

A first scaling analysis of ISSM has been conducted in Larour et al. (2012) presenting the efficiency of the stress balance module up to 180 compute cores for a case of the GrIS using the direct solver MUMPS. In our study, we extend the performance

analysis of ISSM to cover the entire code base using a high-resolution model setup as well as several lower resolution ones executed on 96 to 6144 cores. Since direct linear solvers are rather uncommon for sparse problems of such size, we rely on an iterative solver – based on Habbal et al. (2017) we selected GMRES preconditioned with block Jacobi method.

In addition, in order to be able to pinpoint the effect of different models on the overall performance, we develop an instrumentation scheme that provides detailed performance information closely related to an Earth systems scientist's view of the

code. The challenge here is to develop a setup that does not introduce much overhead, as otherwise the results of instrumented runs would not be representative for the original code base. As a result, then, such an analysis can become part of the standard production environment, providing insight into the code's performance as algorithms or the code's environment change. To that end, a setup is developed that limits the instrumentation overhead through careful filtering, but can leave the code base untouched.

The paper is structured as follows: We start by introducing the underlying physical model and the employed test setup applied to GrIS (Section 2). We then present the software design of ISSM, its parallelization scheme and how it supports efficient simulations (Section 3). The main part of this study is the measurement of the run-time. We present our low overhead sustainable performance measurement instrumentation in Section 4 and the results are presented in Section 5. We continue by

---

[1]formerly Albany/FELIX





discussing the results, including the options for further code optimisation, and complete the study with a discussion from the
prospective of interactive ice sheets, exchanging variables with atmosphere and ocean models rather than just ingesting data
from them, integrated in Earth System Models (ESMs) (Section 6).

## 2 Description of the model and the experiment environment

### 2.1 Description of the model

For this study we focus on a selected subset of the capabilities of ISSM, e.g. we employ only the HO approximation of the
stress balance. This approximation is currently used in ice sheet projections and, in terms of the ice sheet code run as a part
of a fully coupled ESM, it is the most comprehensive level of physics that we expect to be practical in ESMs. We also do
not incorporate other sophisticated modules, such as subglacial hydrology, or advanced approaches for computation of surface
mass balance (SMB). The mathematical model is given in detail in the Appendix Section A1. For a more exhaustive model
description the reader is referred to Rückamp et al. (2019).
The mathematical model for the different modules (Section A1, Eq. A2-A23) are discretized using the finite element method
on an unstructured mesh that is fixed in time. The computation within a time step is conducted in a sequence of different
modules (see Figure 1 for a schematics of the main execution substeps), which means that the different balance equations for
momentum, mass and energy are not solved in a coupled fashion. Furthermore, the stress balance module solves a nonlinear
PDE system using a fixed-point Picard iteration whose each step involves solution of a linear equation system. The total
number of degrees of freedom (DOF), the quantity that determines the size of the matrix for the linear equation systems, differs
substantially between different modules. The fields for velocity, enthalpy, ice thickness, and ice level-set are computed at each
vertex of the mesh using piecewise-linear (P1) finite elements. The enthalpy equation is stabilised with ASUPG (anisotropic
streamline upwind Petrov-Galerkin method Rückamp et al., 2020b).

### 2.2 Description of the benchmark problem

The results presented in this study are based on a realistic setup of the GrIS. The setup was previously used for future projections
(Rückamp et al., 2019) and is utilized here with slight modifications. The bed topography is BedMachine v3 (Morlighem et al.,
2017). The ice thermodynamics are computed with an enthalpy scheme (Section A1.4) initialised with the enthalpy field from
a paleo-climate spin-up similar to Rückamp et al. (2019). Surface topography and 3D velocity are also taken from this spin-up.
Grounding line evolution (Eq. A22) uses the sub-element parameterization of the friction coefficient (Seroussi et al., 2014).
Climate forcing fields are not read in at each time step in the surface mass balance (SMB) module.
The main difference to Rückamp et al. (2019) is that the moving front module is enabled during the scalability tests to capture
its performance. The calving front motion is solved by a level-set method (Eq. A23), which tracks the ice front according to
a kinematic calving front condition (Bondzio et al., 2016, 2017). Here we assume a moving front configuration where the





**Table 1.** Overview of problem setups used in our study. Note that (a) the vertices and elements of the 3D mesh describe the full mesh (i.e. ice-covered and non-ice covered elements), (b) the minimum DOF are the DOF from the 2D mass transport module and (c) the maximum DOF are the DOF from the 3D stress balance horizontal module.

| Name | resolution | time step size | 3D vertices | 3D elements | minimum DOF | maximum DOF |
|---|---|---|---|---|---|---|
| G4000 | 4-10 km | 0.05 years | 566 280 | 1 055 572 | 31 468 | 944 040 |
| G1000 | 1-10 km | 0.0025 years | 2 081 760 | 3 884 468 | 122 417 | 3 672 510 |
| G500 | 0.5-10 km | 0.0001 years | 5 509 080 | 10 282 132 | 335 117 | 10 053 480 |
| G250 | 0.25-10 km | 0.0001 years | 17 111 325 | 31 939 656 | 1 064 669 | 31 940 010 |

advance of the calving from (i.e. by glacier velocity) is compensated by the sum of calving and frontal melting. These settings
basically specify a system close to an equilibrium.

Each horizontal mesh is generated with a higher resolution denoted by $\mathrm{RES_{high}}$ in fast-flowing regions (initial ice surface velocity $> 300\,\mathrm{m\,a^{-1}}$) while maintaining a relatively low, $\mathrm{RES_{low}}$, resolution in the interior. This statically adaptive mesh generation strategy allows for a variable resolution and gives more computational resources to regions of dynamic complexity. Simulation measurements are performed on five meshes (Figure A1, Table 1) using 15 vertical sigma layers refined towards the
base where vertical shearing becomes more important. The meshes differ by employing five different horizontal grid resolutions with $\mathrm{RES_{high}}$ as specified in Table 1. The highest resolution, G250, is ca. 150 m, which corresponds to the resolution of the bed topography dataset used in our study.

In order to measure the performance, we conduct thirty time steps in each run, but we only measure time steps from eleven to thirty. Since we allow in each time step the individual modules to reach their convergence criteria, we intentionally exclude
the timings from a cold-start based on a poor initial guess.

The convergence criteria (see Appendix) for the linear iteration of the stress balance is $\epsilon_1 = 10^{-4}, \epsilon_2 = 10^{-2}$ (no $\epsilon_3$ set). The thermal solver is using $\epsilon_2 = 10^{-3}$. Stress balance and thermal modules are set to use at most 100 nonlinear iterations, but this limit is never reached in all cases observed.

## 2.3 Experimental environment

All experiments are conducted on dedicated compute nodes of the Lichtenberg HPC system with two 48-core Intel Xeon Platinum 9242 and 384 GB of main memory each, connected with an InfiniBand HDR100 network. We employ 48 MPI processes on each node pinned to NUMA nodes. More processes per node are not possible, because of memory limitations of the high resolution model G250. Each experiment runs 3 repetitions, and results fall within a standard deviation of 10%. The basis for our instrumentation is the latest ISSM public release 4.18, which is is compiled with GCC 10.2 (optimization level
-O2), Open MPI 4.0.5 (Graham et al., 2006) and PETSc 3.14 (Balay et al., 2021b).For profiling and tracing we use Score-P 7.0 (Knüpfer et al., 2012) and for data analysis Cube GUI 4.6 (Geimer et al., 2007) and Vampir 9.10 (Nagel et al., 1996).



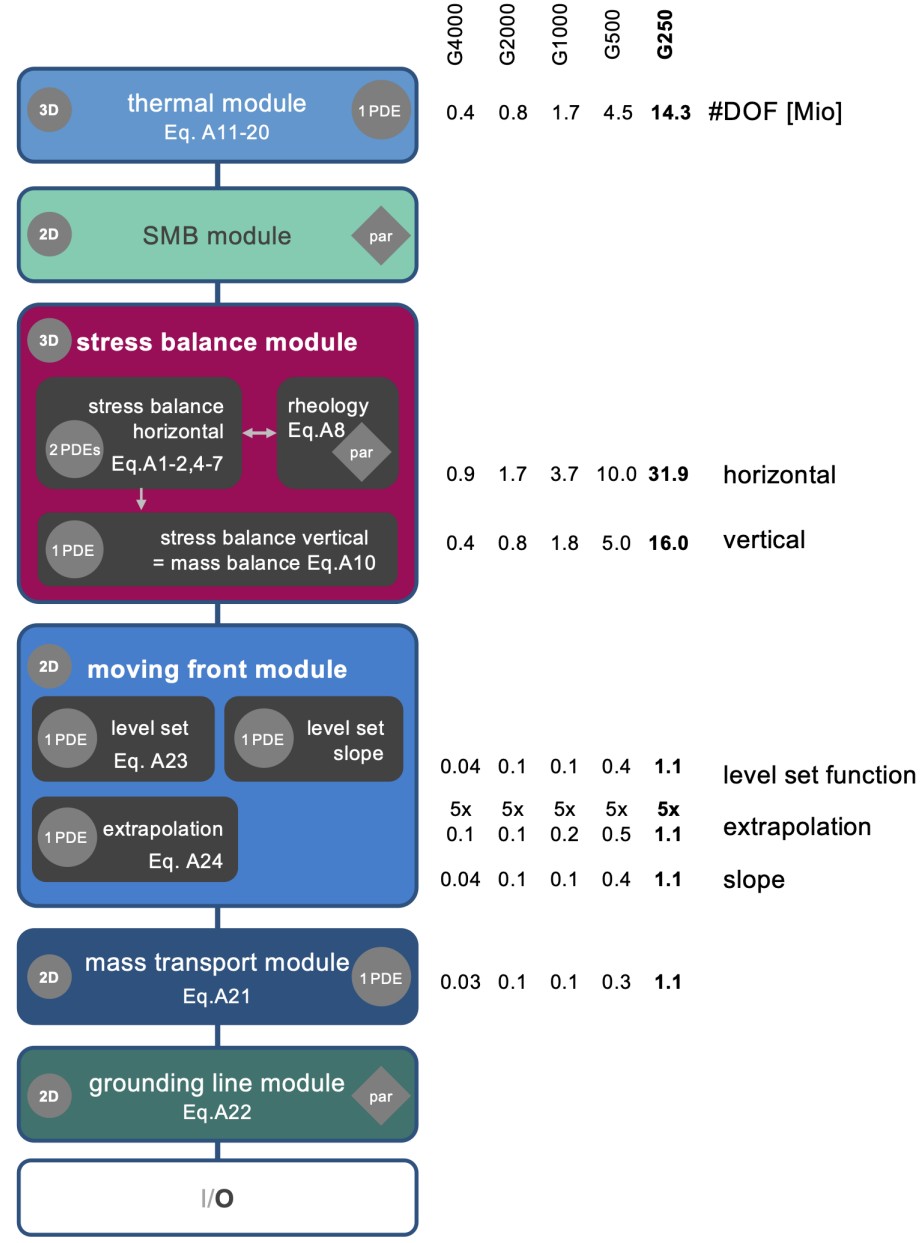

**Figure 1.** The sequence of modules in a transient time step in ISSM. Small grey circles indicate the dimension of the equations of the module (3D, 2D). Larger grey circles with PDE are denoting if and how many partial differential equations are solved. Diamonds with 'par' indicate that only an algebraic equation is evaluated.





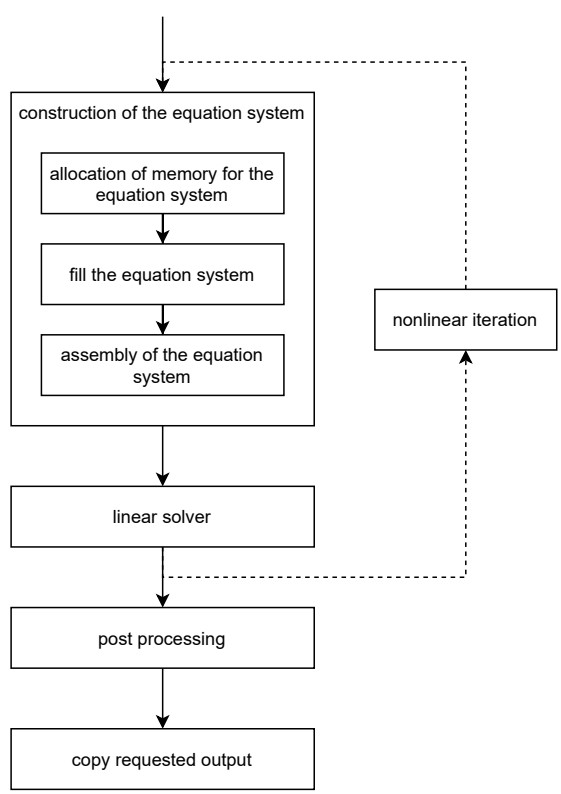

**Figure 2.** Sketch of a solution sequence of ISSM

## 3  Software Design of ISSM

ISSM is implemented in multiple modules, which run in a predefined sequence illustrated in Figure 1 for the transient solution in our GrIS setup. While these modules all utilize the same 2D mesh, the same vertical layer structure and the same data

distribution, they strongly differ in the number of DOF ranging from a large linear equation system for the velocity field in three dimensions to the comparatively low-cost computations needed for two dimensional fields in the mass transport module (the equations solved are given in the Appendix Section A). The solution procedure in each module belongs to one of three different types: The SMB module and the grounding line module calculate an algebraic solution, the mass transport module and the moving front module solve linear PDEs and the thermal module and the stress balance module solve nonlinear PDEs.

The file output for the results of the simulation is carried out in the last step of the sequence.





Although these modules solve different mathematical problems, the modular software designs enables a similar structure for the implementation of the linear and nonlinear equations. This generalized form of the solution sequence of ISSM is shown in Figure 2.

The first step of each sequence step that involves the solution of a PDE is identical and consists of constructing the equation system. Within this step, memory is allocated, the entries of the system matrix are filled in on each mesh element, and the global matrix is assembled. The main difference between the modules lies in how the equation matrix is filled. Here the code iterates over all elements of the mesh and computes an element matrix, whose entries depend on the PDE being discretized. The element entries are then assembled into a global matrix. Next, the equation system is solved. If a module contains a nonlinear iteration – this is the case for the horizontal stress balance and the thermal module – in each step of the nonlinear

solver material properties or basal constraints are updated, and the global linear equation system is solved in the same fashion as for linear PDEs. The nonlinear iteration is repeated until the convergence criterion is reached. Subsequently, the results are post-processed, and the geometry of the mesh is updated as needed. Finally, the requested file output is selected.

While running multiple modules in parallel is not possible due to data dependencies, ISSM parallelizes the solution sequence of individual modules. For this it uses an even distribution of the elements, which is constant over time and independent of

the modules, and each modules handles the parallelization in the same way. During the construction of the equation system memory is allocated locally. Subsequently the element matrices are computed and the equation system is filled. While this step is free of MPI communication, all entries which are assigned to other MPI processes are communicated in the assembly of the equation system leading to many MPI calls. Afterwards the parallel linear solver solves the equation system and the solution is distributed to all MPI processes in the post-processing. In case of a nonlinear system, a convergence criteria has

to be computed, which mainly consists of a single MPI allreduce, and the nonlinear iteration has to be updated potentially involving additional MPI calls. In the final selection of the requested output, data is stored in vectors or reduced to scalars, both operations lead to MPI communication.

## 4 Sustainable Performance Measurement

Large code bases such as ISSM are developed and used over decades. The environment in which they are executed, on the other hand, i.e., the hardware, the operating systems, underlying libraries and compilers, is in a constant state of flux. As a consequence, code development needs not only to address the representation of physics but also to account for these changing operating environments, in particular, the increase in the number of compute cores. Modernizing a code or porting it from one operating environment to the other is likely to affect overall performance. In particular, modules that do not play a significant

role with respect to compute time with a low core count may end up taking a significantly larger chunk of compute time on a larger parallel system and thereby significantly affect the overall performance and scalability. In addition, a code like ISSM is never in a final state: The development of new modules, the implementation of new algorithms in existing modules, or an update of used libraries can have a substantial (positive or negative) performance impact. As a consequence, a continuous performance





monitoring of ISSM is an essential feature allowing to assess the performance on the shifting computational ground the code

lives on. For this reason, the code version of ISSM that we started out with had a basic timing setup to monitor the performance of the 8 modules accounted for in Figure 3 as well as of the setup stage and the linear solver.

As will be shown in the next section, we need to dig deeper to develop a sufficient understanding of the performance behavior of ISSM. To gather this information, we developed a sustainable performance measurement environment which provides performance information that correlates with the algorithmic view of domain scientists. Sustainability here includes

three main factors: (1) The instrumented code needs to be easy to build and use, (2) the instrumentation results need to refer to identifiable modules in the code and (3) measurements must not lead to a significant computational overhead, as it would distort results.

Profiling information for a code can be created in two different ways: Sampling and/or instrumentation. With sampling-based tools, the execution is interrupted at regular intervals and that state recorded (e.g. with HPC-Toolkit, cf. Tallent et al. (2009)),

whereas with the instrumentation approach, calls to monitoring functions are inserted into the source code (or an intermediate representation) manually or automatically. Instrumentation has the advantage that it measures exactly the same regions in each run, thereby improving comparability of results. The main potential disadvantage of instrumentation over sampling is the overhead resulting from too many logging calls, which can be avoided by efficient filtering.

In our work, we use the Score-P instrumentation tool (Knüpfer et al., 2012), whose output in the cubex format is also used

by a variety of other analysis and visualization tools. These cubex files can be used in subsequent tools like Cube GUI (Geimer et al., 2007), Vampir (Nagel et al., 1996), Scalasca (Geimer et al., 2010) or Extra-P (Calotoiu et al., 2013) to gain performance information, provide visualizations or hints for improvement potential.

In this work, we go beyond the instrumentation of the top-level modules and develop an instrumentation that enables an in-depth analysis of ISSM behavior which is closely tied to the algorithmic view of domain scientists through making judicious

use of the features provided by Score-P. Score-P generates profiles and traces based on compiler instrumentation supporting filtering and manually defined user regions. Additionally, Score-P hooks into the PMPI interface (the MPI profiling interface) of the MPI library and is therefore able to track each MPI call with little overhead. This is important as calls to MPI functions are likely causes of synchronisation overhead that we might encounter. In addition, Score-P is able to generate similarly instrumented interfaces for user-defined libraries, which we employ to wrap calls to the PETSc library. Our timing profile thus

includes every PETSc call. This is beneficial since the PETSc calls provide much more context information than MPI calls by themselves. So the profile contains the information whether an MPI call belongs to an assembly, the solver or some other PETSc algorithm, for example.

In order to develop a low-overhead instrumentation, we start out with a full instrumentation (which is generated automatically without effort on our part) and then analyze which modules account for a significant chunk of the run-time. Repeating this

several times and taking the modular structure of ISSM (see Section 3) as a guideline then provides us a structure for efficient instrumentation. In the end, we provide Score-P with a whitelist of 52 functions to instrument and instrument by hand 6 code regions by bracketing them with Score-P instrumentation calls. In the following, we refer, for simplicity, to 58 instrumented





**Table 2.** Details of our filter, the final profile and the overhead of the measurement environment 30 timesteps of G250 with 3072 MPI processes

| instrumented regions (IR) | number of IR | number of calls of IR | additional calls of IR |
|---|---|---|---|
| MPI | 33 | 14 642 billion | |
| MPI & PETSc | 89 | 14 726 billion | 84 024 million |
| MPI & PETSc & ISSM (filtered) | 147 | 14 726 billion | 57 million |

regions (IR). In fact, the maintenance of an instrumented version of ISSM would be easier if the 6 regions were functions, as, in that case, no Score-P instrumentation code would need to be inserted in ISSM.

These 58 functions cover the hot paths of the main parts of the code. We mention that this process of finding the hot paths of a code has since been (partially) automated with the PIRA tool (Arzt et al. (2021)). When we started this work, PIRA was not yet able to deal with a code base such as ISSM.

Since we paid attention to include all functions and methods which are on the call path to a function we instrument, in order to get the context of each measured region, our function white list includes the entry point of each physics module (which have

also been measured by ISSM internal timings), the calls of the individual solution sequences, and the top level calls of the logical steps of the algorithms, e.g. allocation of memory, computation of element matrices, assemble of matrices and vectors and the linear solver.

Table 2 illustrates the impact of instrumentation for a model run for 30 steps with the G250 resolution on 3072 MPI processes. In addition to the 58 ISSM functions whose instrumentation was triggered through filtering or bracketing of regions, we see that

56 different functions of PETSc and 33 different MPI function are called and measured. The number of calls to instrumented functions of ISSM is quite low (57 million), the number of PETSc functions called (84 billion) is roughly 1500 times higher, which is far exceeded by the number of MPI calls.

But even with this high number of calls to MPI functions, the instrumentation overhead remains low. The profiling of the 15 trillion MPI calls results in a run-time overhead of about 2.5%. When, in addition, the 84 billion PETSc functions

are instrumented, we do not detect any noteworthy additional overhead, the same holds for the addition of the 58 million calls related to ISSM. On the other hand, a fully automatic instrumentation of ISSM results in an overhead of over 13 000 % according to our measurements on a coarser grid. Fully instrumented binaries are not executable in reasonable time, and any performance evaluations made on their basis may have little relevance with respect to the original code.

The bottom line is that our instrumentation scheme keeps overhead low, even for large-scale runs, thus ensuring that the

measured code is representative of the original source. As a result, it is quite feasible to periodically run an instrumented version of the code as part of the regular work of domain scientists, as a safeguard against surprises arising, for example, from a changed MPI library.

Through the performance analysis of ISSM we recognized, for example, that the matrix of the equation system in the stress balance horizontal module is being reallocated in each iteration of the nonlinear iteration scheme. Since the structure of the





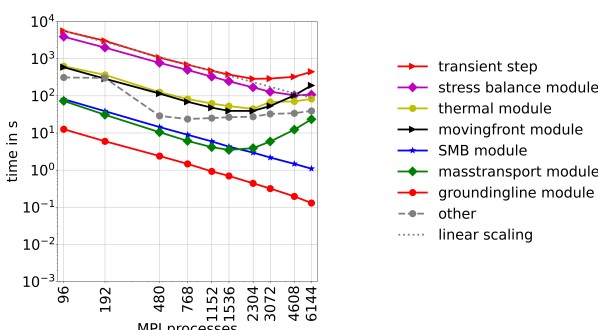

**Figure 3.** Run-time of a transient time step of ISSM without output handling in resolution G250 (draft)

matrix does not change during these iterations, we modified the code to preallocate the matrices. Reusing them saved substantial time in the allocation and assembly of the equation system of the stress balance horizontal module. For example, running with 3072 cores, the time for matrix allocation was reduced by 91%, the time for matrix assembly was reduced by 85%, and the overall run-time of stress balance horizontal module by 31%. The performance numbers shown in the following section are based on this optimized version.

Preallocation is not an option for the thermal module due to the physics of the system, which is why it is not preallocated. Since the nonlinear iteration scheme of the thermal module only needs one to three nonlinear iterations in our setup, the reallocation does not matter as much as in the stress balance horizontal module anyway.

## 5 Scalability of ISSM

In this section we present the results of the measurements for the entire transient time step (Figure 3) as well as for individual 255 modules (Figure 4-7). Output handling was measured and manually excluded in the post processing of the profiles and thus is not reflected in the run-times shown. We focus on the results for the highest resolution G250 first and compare the throughput of different resolutions afterwards. The number of MPI processes (and the number of DOF) is plotted on the x axis and the y axis shows the the run-time in seconds. Due to the log scale of both axis linear scaling is represented by straight lines. All plots employ the same scale for the y axis for easier comparison.

### 5.1 Transient step

Figure 3 displays the total run-time for 20 time steps and a breakdown to the individual modules for the transient step. From 2304 MPI processes on, the transient step deviates from linear scaling and, most importantly, the computational costs start to rise from 3072 MPI processes onward.

However, the stress balance module scales linearly up to 3072 MPI processes and reasonably above that. Although the 265 stress balance module does not scale linearly, its run-time is still monotonously declining over the number of employed MPI





processes. In contrast, the earlier increase (i.e. at a lower number of MPI processes) in the run-time of the thermal, mass transport and moving front modules is more prominent. Since they scale worse than the transient solution, they become more relevant with increasing numbers of processors. The worst scalability is found for the moving front module, which contributes even more than the stress balance module from 4608 MPI processes on. The main reason for these discrepancies in the scaling

behavior is the fact that the individual modules solve equations with differing numbers of total DOF and different computing costs per element. In the following, we discuss the scaling behaviour and the algorithmic parts which cause it. Modules that do not solve any PDE (SMB and grounding line modules) scale linearly and are not further investigated.

## 5.2  Stress balance module

Since the stress balance module is the most time consuming module of ISSM, it is the most important module with respect to

performance optimization and thus discussed first. Exploiting the natural anisotropy of the problem, the horizontal and vertical components are solved in an uncoupled fashion, and the structure of the PDEs varies greatly between these two components. Therefore, we present them here separately. The run-time of both modules is displayed in Figure 4.

We observe that the stress balance horizontal module is by far more expensive to solve than the vertical, which is expected due to more DOF in the former. The scaling behaviour of both modules differs significantly with the stress balance vertical module

scaling worse. For the horizontal stress balance module, run-time is still monotonously declining at 6144 MPI processes but starts to deviate from the linear scaling. The stress balance vertical module exhibits a minimum run-time at 2304 MPI processes and slows down by about a factor of four with 6144 MPI processes. In the horizontal case, linear scaling breaks down when the DOF per MPI process fall below 10 000, while the vertical case never reaches 10 000 DOF per MPI process.

The execution time of the stress balance horizontal module and the stress balance vertical module on low core counts is

dominated by the costs for the computation of the entries of the matrix, which scales linearly with the number of cores for setups considered in this work. Most notably, the costs for the matrix assembly are in both cases rising from 1152 cores on despite the large difference in the size of the problem. The linear solver does not represent a problem either in the stress balance horizontal module or in stress balance vertical module: It does not need a significant amount of time and its scaling is sufficient. While the stress balance vertical module is solved in a linear equation system, the nonlinear equation system of the

stress balance horizontal module has to be solved iteratively and needs approximately twelve iterations per time step.

## 5.3  Thermal module

The run-time for the thermal module is presented in Figure 5. The linear scaling of the module breaks down with about 12.000 DOF per MPI process and, after reaching a minimum at 2304 MPI processes, the execution time rises again. The execution time is dominated by three components: The computation of matrix system entries, the matrix assembly and the

update of basal constraints. The assembly of the matrix system scales only up to 768 MPI process, which corresponds to about 18 600 DOF per MPI process.

The thermal module contains a nonlinear iteration schema in which the basal boundary conditions are updated. This update is expensive, and the execution time does not change much as the number of cores increases. Furthermore, Figure 5 shows





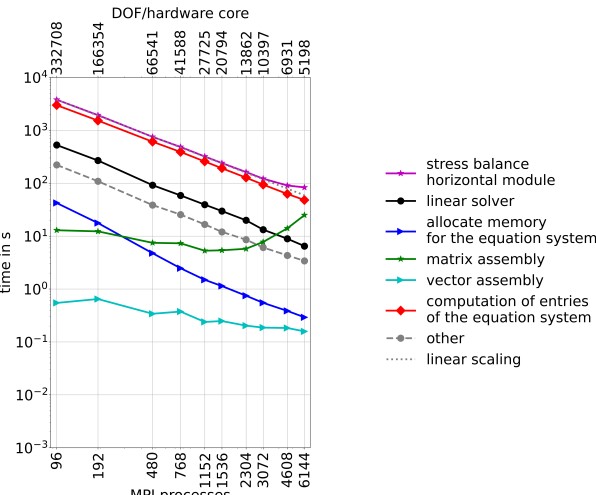

(a) Scalability of the stress balance horizontal module for G250.

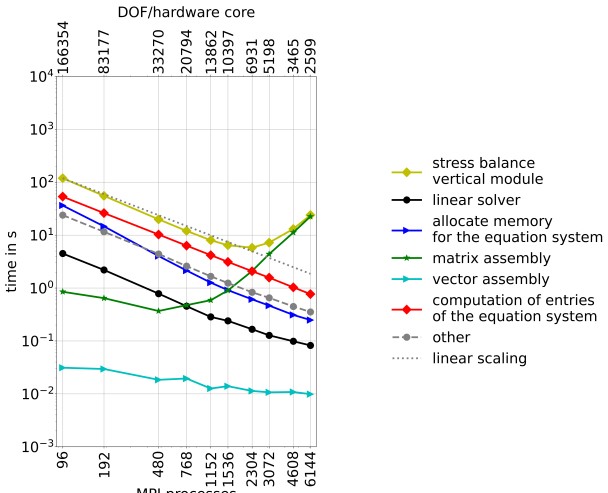

(b) Scalability of the stress balance vertical module for G250.

**Figure 4.** Run-time of the stress balance computation of ISSM Greenland Model G250 (draft).





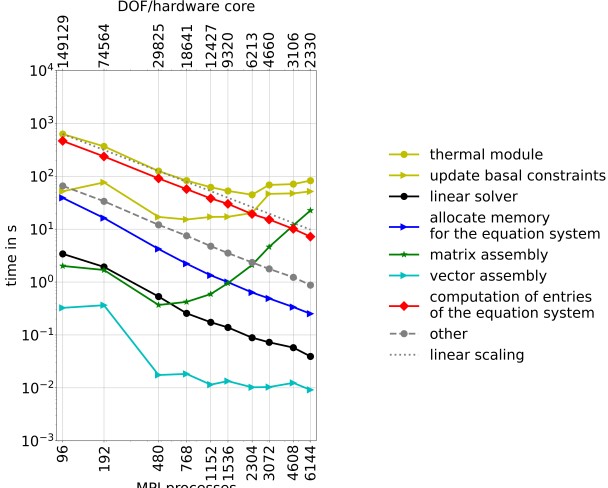

**Figure 5.** Run-time of the thermal module for G250.

that the costs of the linear solver can be neglected. The computation of the convergence criteria and the post-processing of the
results (summarized as 'other') do not need significant run-time and scale linearly within the range of our experiments.

## 5.4   Moving front module

The moving front module consists of three individual modules: (1) a levelset module that is computed first, followed by (2) a
module evaluating the slope of the level set function and lastly (3) the extrapolation module. As shown in Figure 3, the moving
front module becomes as costly as the very expensive stress balance module while slowing down even more than the thermal
module above 3072 MPI processes.

     The run-time of each step is displayed in Figure 6. All three constituent models scale up to 768-1152 MPI processes and
are mainly limited by the assembly of the equation system. Whereas in the extrapolation module the matrix assembly does not
even scale for $\sim 12\,000$ DOF per MPI processes, in the level set and level set slope modules, the costs for the matrix assembly
rise once DOF per MPI processes is below $10\,000$.

The amount of time required for allocation of memory for the level set and level set slope modules is similar to that of stress
balance and thermal modules. In contrast, the allocation of memory is more time consuming for the extrapolation module. This
is likely due to repeatedly solving a diffusion equation that accumulates larger costs.

     The linear solver does only take a significant amount of time in the extrapolation module and scales linearly. The execution
time of the linear solver of the level set and level set slope modules is negligible. Other routines are summarised in the grey
dashed line, but because of an almost linear scaling, they do not play an important role.





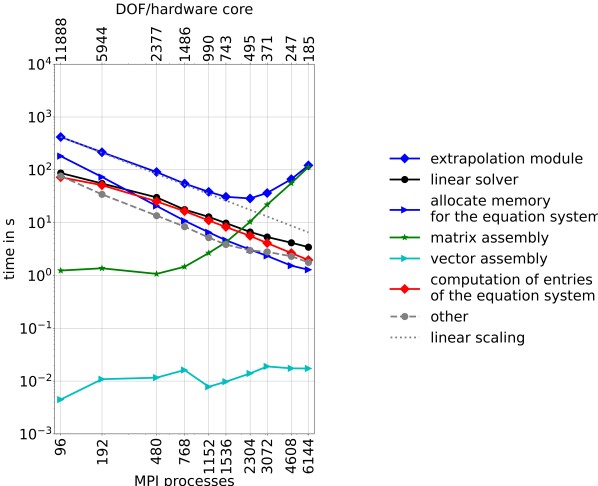

(a) Run-time of the extrapolation module for G250.

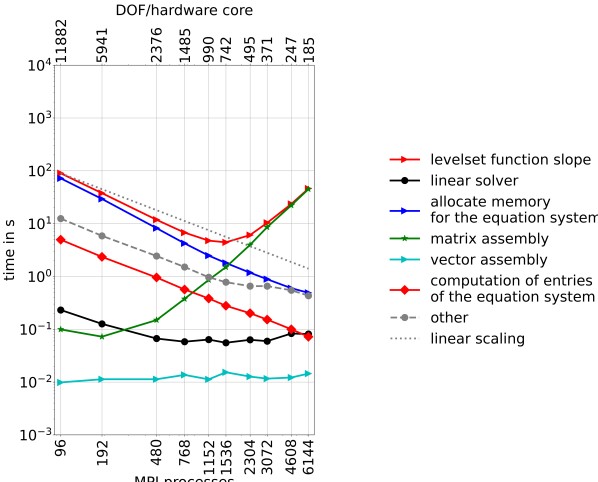

(b) Run-time of the levelset function slope of ISSM for G250.

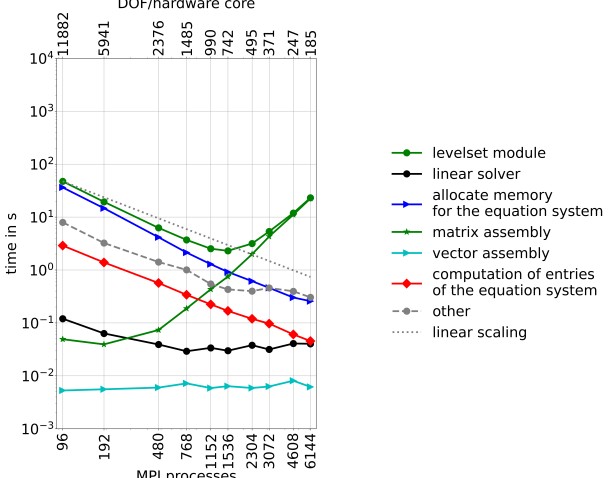

(c) Run-time of the levelset module for G250.

**Figure 6.** Run-time of the moving front module for G250 (draft).





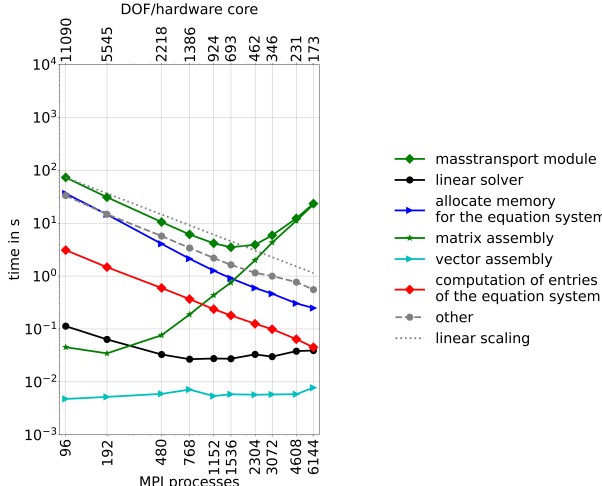

**Figure 7.** Run-time of the mass transport module for G250.

## 5.5 Mass transport module

This module is characterised by a particularly low number of DOF which is 30 times smaller than in the horizontal stress balance and 13 times less than in the thermal module. The overall module scales up to 1536 MPI processes (Figure 7) despite the number of DOF per MPI process being already as low as 693. Figure 7 reveals that matrix assembly is the main scaling problem of the mass transport module once the number of DOF per MPI process drops below about 5500. The linear solver has negligible overall costs; however, its linear scaling ends at the same number of cores (and with the same DOF/core) as the assembly of the matrix system is becoming more costly than the allocation of memory. Other routines – such as averaging over depth and the update of constraints, to name the most expensive ones – scale linearly and are not further investigated at this point.

## 5.6 Throughput of the transient solution

Of high interest in terms of planning simulations is the quantity SYPD, as it reflects the total time needed to conduct a certain simulation. As the typical applications of ice sheet models vary strongly in simulated time periods and resolution, we conducted here simulations with coarse resolutions (G4000), a resolution that is still higher than that used in the current paleo-simulations, as well as with the highest resolution we could afford (G250). We estimated SYPD from our simulations of 20 time steps and scaled them up to one year by also taking into account differing time step sizes for each resolution. As displayed in Figure 8, SYPD is approaching a maximum at 1152-2304 MPI processes with a decline at higher numbers of cores. The coarser the resolution the larger the SYPD. Increasing spatial resolution requires a smaller time step impacting strongly the throughput at high resolutions.



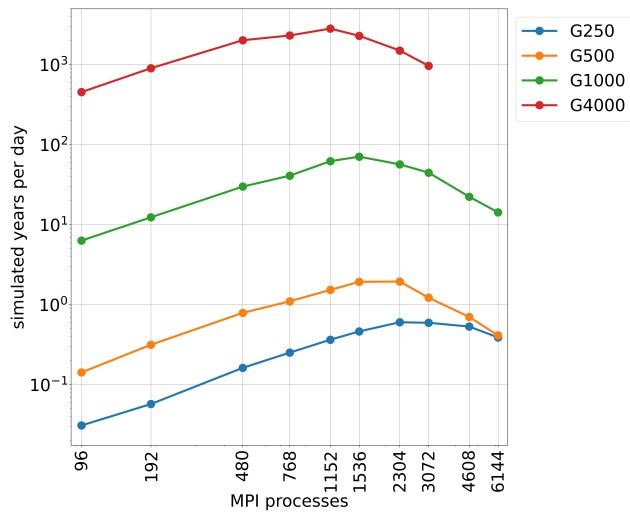

**Figure 8.** SYPD for various grid resolutions estimated from run-time of 20 timesteps.

### 5.7 Scalability comparison for different resolutions

As our ability to assess the scalability of the code might be limited by the size of the problem we solve, we are conducting in this last part a comparison for simulations of Greenland in different resolutions. Figure 9 displays the total run time for five different resolutions. The corresponding DOF per MPI process for the different modules is presented in Figure 1 and an overview of the minimum and maximum number of DOF for each resolution is given in Table 1. There is a factor of about 34 between the minimum, as well as the maximum, DOF betweeen G250 and G4000. The overall behaviour is similar for all

resolutions: the code scales linearly until a threshold and execution time is rising from that minimum on. The minimum in execution time is reached at 2304 MPI processes for the high resolutions G250 and G500. For lower resolution (and lower DOF) the minimum shifts towards lower number of MPI processes. For G4000 the minimum is reached at half of the number of MPI processes than for G250 and G500.

## 6 Discussion

The performance analysis for the components of individual modules reveals that the major scaling issue is the assembly of the equation system matrix as shown in Figures 4-7. Figure 10 displays the costs of matrix assembly in all modules over the DOF per MPI process. It reveals that the execution time for matrix assembly is reaching a plateau (vertical stress balance and extrapolation modules) or potentially approaching a plateau from a certain DOF per MPI processes onwards. The minimum is similar for the 3D cores at about 10 000 DOF per MPI processes, whereas the minimum of 2D modules is below that value. The

range of costs for matrix assembly is varying for 2D modules by a factor of 104–614, while it is for 3D modules only a factor of 5-60 with the minimum of five being found for the vertical stress balance module. The matrix assembly costs for the two



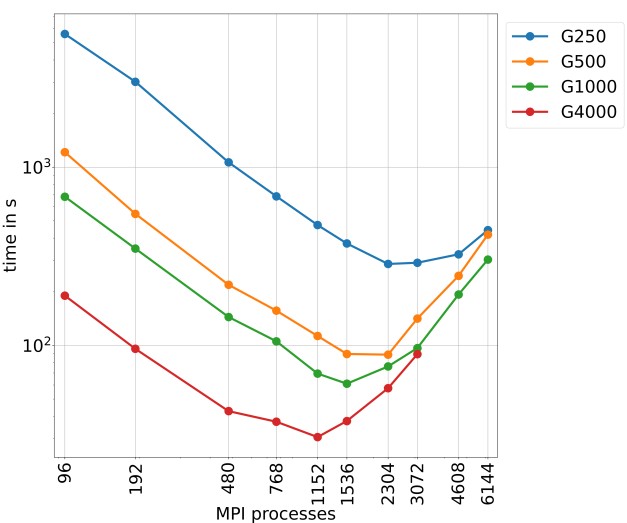

**Figure 9.** Run-time for various grid resolutions.

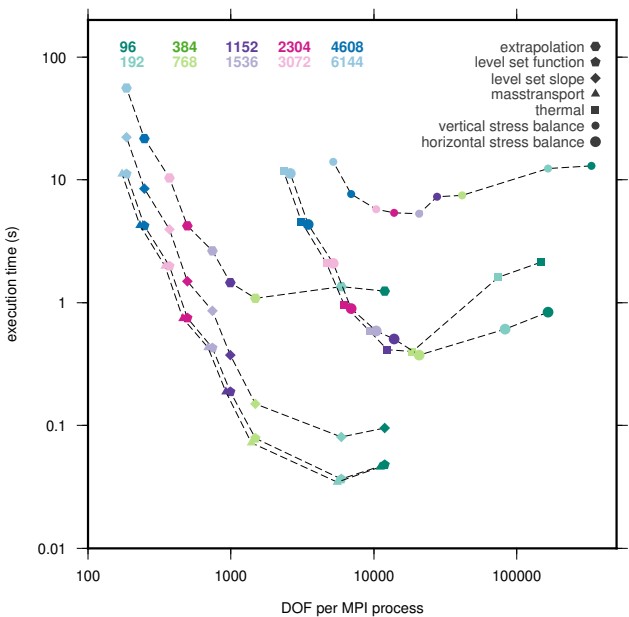

**Figure 10.** Execution time for matrix assembly for various modules. Symbols are representing the modules, while the color is denoting the number of MPI processes.



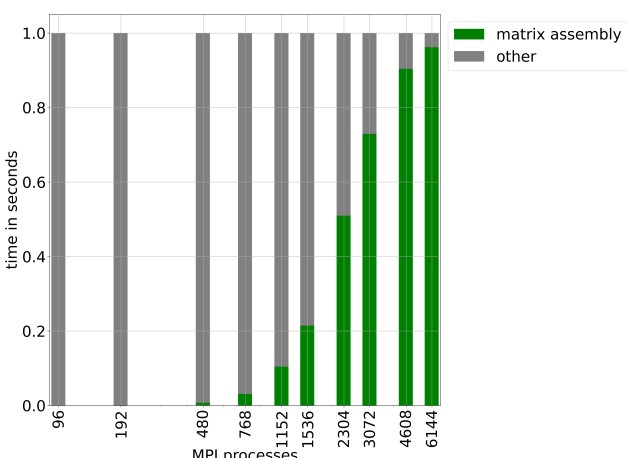

**Figure 11.** Percentage of computation time spent in matrix assembly versus the remaining computations in mass transport module.

modules with nonlinear iterations (horizontal stress balance and thermal modules) scale for larger numbers of MPI processes very similarly to each other.

Matrix assembly consists almost entirely of MPI communication, and the cost of communication increases from a certain
point on as the number of MPI processes grows. The allocation of memory and the computation of entries of the equation system, on the other hand, are both core-local computations and hence scale linearly with the number of cores. In particular, this part of the computation will scale further, because elements are distributed evenly on MPI processes, no matter how many or few elements are computed on each core. In all these modules, the run-time contribution of vector assembly is insignificant and the linear solver is either insignificant or scales well.

Our measurements reveal that the scalability is breaking down in all modules by poor performance of the matrix assembly. The tipping point mainly depends on the costs of the allocation of memory and the time for computing the element entries of an individual PDE, the amount of elements per MPI process, and the data locality of the assembly, which impacts the amount of inter-core communication required for the assembly. So, for a fixed core count, increasing DOF increases performance but, with an increasing number of MPI processes, the communication overhead of the assembly starts dominating at some point.
Similar behaviour was also found in other studies (e.g. Perlin et al., 2016). This is also clearly reflected in Figure 11 which shows the increasing run-time of mat assembly as the number of cores increases.

While not in the focus of the current study, we also noticed that the output routine does not scale well, either. The main reason for this is an indexing scheme tailored towards post-processing in the MATLAB/python user interface which does not exhibit good data locality and leads to a large number of MPI calls. Here, we suggest the more data-local indexing scheme already
used in the creation of the equation system. The compatibility to the user interface could then be achieved in a reordering post-processing of the results, which can be done trivially parallel by distributing different output vectors and different time steps among available cores.



The performance analysis of an ice sheet code needs to keep the challenging numerical underpinnings of the problem in mind: The code solves in a sequential fashion a number of different modules, each of which solves an equation system of a
very different type and with a different number of DOF. Therefore this type of code is inherently prone to the situation that a particular domain decomposition may be optimal for the module with the largest number of DOF and scale well with increasing core counts, while the performance of the module with a smaller number of DOF may not experience any improvement – or even worsen – with increasing core counts.

Also other components of ESMs are facing the issue that modules with fewer DOF are limiting scalability. For the finite
volume sea ice-ocean model FESOM2, similar scalability issues were found for 2D computations (Koldunov et al., 2019) such as the sea ice module and the linear solver. For FESOM2, the problem boils down to a large number of short but too frequent communications that were suggested to be addressed by widening the halo layer.

Our analysis reveals that the migration of lateral margins is becoming costly with increasing number of cores. This module includes the extrapolation of some solution fields performed via solving a diffusion equation over the ice-free regions
constrained by the values calculated in the ice-covered region. This approach generates the smallest stiffness matrices in the overall sequence because ISSM treats Dirichlet boundary conditions with a lifting method so that only the free degrees of freedom are included in the stiffness matrix. Since all vertices located on ice are constrained, the free degrees of freedom are only the vertices that are outside of ice, and one ends up with a very small number of DOF per MPI process. One approach to address this problem would be to use an alternative approach for treating Dirichlet boundary conditions such as including
entries for all nodes in the stiffness matrix, setting rows of constrained nodes to 0 except along the diagonal, and changing the right hand side to the value of the constraint.

In addition, wherever a low number of DOF per MPI process is limiting scalability with increasing core counts, the number of DOF could be increased by switching to P2 (quadratic) elements or solving the full Stokes problem. In this context, one must find a reasonable balance between increasing the size of the problem for the sake of making the node computation more
expensive while keeping communication constant, and increasing total computational costs disproportionally with respect to the additional knowledge gain. At this point, it should also be mentioned that the bedrock topography is only insufficiently known, and resolutions finer than 150 m are to date limited by the lack of input data for such simulations.

In order to increase the throughput, new modelling strategies are worth investigating. The nonlinear iterations are contributing to the overall costs of a time step significantly. Depending on the particular application, the number of nonlinear iterations
may be reduced. Thus far, nothing is known about the effect of such a reduction, and care must be taken not to miss abrupt changes in the system. A simulation study comparing the resulting evolution of the temperature field with and without iterative update of the basal constraints can assess this effect. It is here also worth considering employing error indicators to steer the number of nonlinear iterations. Similar to error indicators used for adaptive mesh refinement (dos Santos et al., 2019), physics-based approaches may be particularly attractive.
Future simulation strategies may comprise more on-the-fly analyses than is currently standard in ice sheet codes. So far only few scalars are computed, while an in-depth analysis is conducted in the post-processing. Some analyses may be conducted





on the level of processors, while others need to be run globally. In particular, sensitivity studies for tuning model parameters would benefit from such an on-the-fly analysis with simulations producing unrealistic results quickly terminated.

From the perspective of ice sheet codes running coupled in ESMs, we recommend to consider higher resolutions even if long
time scales are anticipated, as this leads to better scaling than coarse resolution. If the anticipated SYPDs cannot be met, the ice sheet code is to be run on its peak performance, rather than the maximum number of cores available. To this end, investigations of optimal sharing of resources between codes of ESMs should be conducted. Similar to ESMs (Baur et al., 2021), ISSM still falls short in reasonable SYPD for the tasks ahead. The recommendations and directions given by (Baur et al., 2021) thus will need to be adopted to the future development of ISSM too.

By means of the performance analysis we also identified some avenues for future improvement of the code. The reuse of the equation system matrix is also applicable to the multiple executions of the extrapolation step, and a change of indexing in the requested outputs module can be used to improve the memory locality. Furthermore, our instrumentation is suitable for efficient tracing and load imbalance detection. The major load imbalances occur in the computation of matrix entries and lead to load-imbalanced matrix assembly. The matrix assembly is clearly a key computation that warrants further investigation.

Although our performance instrumentation leads to very a modest overhead, this can be further diminished. About 99 % of the MPI calls belong to three functions, MPI_Iprobe, MPI_Test and MPI_Testall, and about 90% of the PETSc calls recorded by Score-P refer to setvalue() routines used in filling buffers for matrix assembly. If the MPI calls are not of interest, they can be disabled in groups via the Score-P interface, or individual functions can be excluded by implementing and preloading a functionless PMPI interface. Currently, the library wrapping interface of Score-P does not allow for the easy exclusion
of certain functions in a library (i.e. a whitelist/blacklist functionality for library functions). However, the overhead of our instrumentation would clearly be reduced further if instrumentation of these low-level PETSc calls could be avoided.

## 7 Conclusions

To analyse the practical throughput of a typical application of ISSM, we conducted transient simulations for the Greenland Ice Sheet in five different horizontal resolutions. We present run-time measurements for individual code modules based on an
instrumentation of the ice sheet code with Score-P. We conclude that ISSM scales up to 3 072 MPI processes in the highest resolution that we tested (G250). While it was expected that the stress balance module would dominate the run-time, we found that simulating the motion of the lateral margins becomes the main cost factor from 4608 MPI processes cores on. We find major scaling challenges due to the assembly of the system matrix, in particular, when the number of DOF per MPI processes is falling below 10 000. The maximum throughput for all horizontal resolutions was reached at 1152-2304 MPI processes and
is particularly small for the highest resolution due to severe time step restrictions.

This study also showed that meaningful in-depth performance analysis of ISSM can be performed at little cost and with minimal code changes, which could be eliminated completely in the future by a very limited refactoring of the code. An instrumented, user-oriented, low-overhead profiling version of ISSM can then be built from the unmodified main source of ISSM. Thus, scientists using ISSM can monitor performance of their code as their computational environment evolves, without





having to worry to about carrying instrumentation code into their new code branch. Future advances in instrumentation with respect to the filtering of library routines could further decrease instrumentation overhead to the level where it is negligible, thereby allowing continuous performance monitoring, which would provide valuable information for the creation of execution models of ISSM and its modules.

*Code and data availability.* ISSM version 4.18 (Larour et al., 2012) is open source and freely available at https://issm.jpl.nasa.gov/ (last
access: August 13, 2021). A copy of the source code including minor changes and scripts to build ISSM on the HHLR is available on https://doi.org/10.48328/tudatalib-613. The Greenland setup is available on https://doi.org/10.48328/tudatalib-614 and the generated profiles are stored at https://doi.org/10.48328/tudatalib-612.

## Appendix A: Appendix

### A1 Mathematical model

Let $\Omega_i(t) \subseteq \mathbb{R}^3$ be a three-dimensional domain with $t \in [0, T]$. All equations are given in Cartesian coordinates, of which $x$ and $y$ are in the horizontal plane and $z$ is parallel to the direction of gravity and positive upwards. Let the following function be given

$$\boldsymbol{v} \colon \mathbb{R}^3 \longmapsto \mathbb{R}^3$$
$$(x, y, z) \longmapsto \boldsymbol{v}(x, y, z)$$
$$:= (v_x(x, y, z), v_y(x, y, z), v_z(x, y, z))$$

with $\boldsymbol{v} \in C^2(\mathbb{R}^3, \mathbb{R}^3)$ being the velocity for the time $t \in [0, T]$. We assume the ice to have constant density $\rho_i \in \mathbb{R}_+$ and hence being incompressible. Let the viscosity being given as $\eta \colon \mathbb{R}^3 \to \mathbb{R}_+$, with $\eta \in C^1(\mathbb{R}^3, \mathbb{R})$ for $t \in [0, T]$. The pressure is given as $p \colon \mathbb{R}^3 \to \mathbb{R}$, with $p \in C^1(\mathbb{R}^3, \mathbb{R})$ for $t \in [0, T]$. The temperature is given as $T \colon \mathbb{R}^3 \to \mathbb{R}$, with $T \in C^1(\mathbb{R}^3, \mathbb{R})$ for $t \in [0, T]$, similarly the enthalpy is $E \colon \mathbb{R}^3 \to \mathbb{R}$, with $E \in C^1(\mathbb{R}^3, \mathbb{R})$ for $t \in [0, T]$. Let the ice thickness being given as $H \colon \mathbb{R}^2 \to \mathbb{R}$, with $H \in C^1(\mathbb{R}^2, \mathbb{R})$ for $t \in [0, T]$. The normal vector $\boldsymbol{n} \colon \Gamma \times [0, T] \to \mathbb{R}^3$ be the normal vector field on $\partial\Omega = \Gamma_s \cup \Gamma_b \cup \Gamma_{cf}$ pointing
out of the ice body by convention. The boundary at the ice-atmosphere transition is denoted by $\Gamma_s$, the ice-bed interface $\Gamma_b$ and the calving front as $\Gamma_{cf}$.

### A1.1 Momentum balance

The momentum balance used in this study is the Blatter-Pattyn higher-order approximation (Blatter, 1995; Pattyn, 2003). This approximation is reducing the Stokes equation to two PDEs for the horizontal velocities $v_x, v_y$ by neglecting the bridging



stresses and assuming the vertical component of the momentum balance to be hydrostatic.

$$\frac{\partial}{\partial x}\left(4\eta\frac{\partial v_x}{\partial x}+2\eta\frac{\partial v_y}{\partial y}\right)+\frac{\partial}{\partial y}\left(\eta\frac{\partial v_x}{\partial y}+\eta\frac{\partial v_y}{\partial x}\right)+\frac{\partial}{\partial z}\left(\eta\frac{\partial v_x}{\partial z}\right)=\rho_i g\frac{\partial h_s}{\partial x} \tag{A1}$$

$$\frac{\partial}{\partial x}\left(\eta\frac{\partial v_x}{\partial y}+\eta\frac{\partial v_y}{\partial y}\right)+\frac{\partial}{\partial y}\left(4\eta\frac{\partial v_y}{\partial y}+2\eta\frac{\partial v_x}{\partial x}\right)+\frac{\partial}{\partial z}\left(\eta\frac{\partial v_y}{\partial z}\right)=\rho_i g\frac{\partial h_s}{\partial y} \tag{A2}$$

where $h_s$ is the surface topography of the ice sheet, $\rho_i$ the ice density, and $g$ the gravitational acceleration. From the vertical component of the momentum balance, we find the pressure in HO to be

$$p = \mathrm{t}^D_{xx} + \mathrm{t}^D_{yy} - \rho_i g(h_s - z) \tag{A3}$$

with the normal deviatoric stresses $\mathrm{t}^D_{xx}, \mathrm{t}^D_{yy}$ in $x$ and $y$ direction respectively. They are evaluated from the constitutive relation $\mathrm{t}^D_{ij} = 2\eta D_{ij}$ with $D_{ij}$ the strain-rate tensor components.

The boundary condition of the momentum balance is at all boundaries $\Gamma_s, \Gamma_b, \Gamma_{cf}$ a stress boundary condition, but of different kind. The ice surface $\Gamma_s$ is treated as traction free. At the ice base $\Gamma_b$ the kinematic boundary condition of the momentum balance is given as the no penetration condition $\boldsymbol{v}\cdot\boldsymbol{n} = BMB$ with $\boldsymbol{n}$ reducing in HO to a normal vector in $z$-direction only. The stress boundary condition in tangential direction is given by the friction law that is implemented in terms of basal stress, thus $\boldsymbol{\tau}_{\mathrm{b}} = f(\boldsymbol{v}_{\mathrm{b}})$ as

$$(\mathbf{t}\cdot\boldsymbol{n})\cdot\boldsymbol{t}_1 = \tau_{\mathrm{b},1} \tag{A4}$$

$$(\mathbf{t}\cdot\boldsymbol{n})\cdot\boldsymbol{t}_2 = \tau_{\mathrm{b},2} \tag{A5}$$

with the tangential vectors $\boldsymbol{t}_1, \boldsymbol{t}_2$ and the basal drag $\boldsymbol{\tau}_{\mathrm{b}} = (\tau_{\mathrm{b},1}, \tau_{\mathrm{b},2})$. The basal velocity $\boldsymbol{v}_{\mathrm{b}}^{\parallel} = (v_{\mathrm{b},1}, v_{\mathrm{b},2})$ is then determined by a friction law. Basal drag and velocity are two-dimensional vectors acting in the basal tangential plane. We use a Weertman-type friction law with submelt sliding

$$\boldsymbol{\tau}_{\mathrm{b}} = -C_b^{-\tilde{p}}\,\boldsymbol{v}_{\mathrm{b}}^{\parallel}|\boldsymbol{v}_{\mathrm{b}}^{\parallel}|^{\tilde{p}-1}\,N^{\tilde{q}/\tilde{p}}\,e^{-T_b'/(\tilde{p}\gamma)} \tag{A6}$$

reformulated as a Neumann boundary condition (Rückamp et al., 2019) with $C_b$ the drag coefficient, $T_b'$ the basal temperature relative to pressure melting point and $\gamma, \tilde{p}, \tilde{q}$ parameters given in Table A1.

Finally, at the (vertical) calving front $\Gamma_{cf}$ the boundary condition is

$$(\mathbf{t}\cdot\boldsymbol{n})\cdot\boldsymbol{n} = \begin{cases} 0 & z \geq z_{sl} \\ -\rho_{\mathrm{oc}}gz & z < z_{sl} \end{cases}, \tag{A7}$$

with $z_{sl}$ the elevation of the sea level.

### A1.2 Rheology

The viscous rheology of ice is treated with a regularized Glen flow law (Eq. A8), a temperature-dependent rate factor for cold ice, and a water-content-dependent rate factor for temperate ice.

$$\eta = \frac{1}{2}A(T,p,W)^{-1/n}E_i^{-1/n}d_e^{(1-n)/n} \tag{A8}$$



with the creep exponent $n = 3$, the flow rate factor $A$ depending on temperature and water content $W$, the enhancement factor $E_i$ and $d_e$ the second invariant of the strain-rate tensor $D_{ij}$.

### A1.3 Mass balance

The ice is treated as an incompressible material and hence the mass balance reduces to $\operatorname{div} \boldsymbol{v} = 0$, which is used in HO to derive the vertical velocity

$$v_z = v_z|_{z=h_\mathrm{b}} - \int\limits_{h_\mathrm{b}}^{z} \left( \frac{\partial v_x}{\partial x} + \frac{\partial v_y}{\partial y} \right) dz' \tag{A9}$$

with $v_z|_{z=h_\mathrm{b}}$ the vertical velocity at the ice base $h_b(x,y)$. This quantity is given by the kinematic boundary condition at the ice base that is assumed to be quasi-static

$$v_z|_{z=h_\mathrm{b}} = v_x \frac{\partial h_b}{\partial x} + v_y \frac{\partial h_b}{\partial y} - \mathrm{BMB} \tag{A10}$$

with BMB the basal melt rate.

### A1.4 Enthalpy balance

We solve the enthalpy balance equation to resolve cold-, temperate-, or polythermal-ice states (Eq. A11, Aschwanden et al., 2013; Kleiner et al., 2015). The evolution equation for enthalpy $E$ reads as

$$\rho_i \left( \frac{\partial E}{\partial t} + \boldsymbol{v} \cdot \nabla E \right) = -\nabla \cdot \boldsymbol{q} + \Psi, \tag{A11}$$

with $\boldsymbol{q}$ the heat flux and $\Psi$ the source term. The advection dominated problem is stabilized with the anisotropic streamline upwind Petrov–Galerkin (ASUPG) method (Rückamp et al., 2020b). The heat flux is given by

$$\boldsymbol{q} = -K_{\mathrm{eff}} \nabla E = - \begin{cases} K_c \nabla E & E < E_{\mathrm{pmp}} \\ K_0 \nabla E & E \geq E_{\mathrm{pmp}} \end{cases}, \tag{A12}$$

with $K_c = k_i / c_i$, $k_i$ the thermal conductivity and $c_i$ the specific heat capacity of ice. The discontinuous conductivity, $K_{\mathrm{eff}}$ is treated with a geometric mean (Rückamp et al., 2020b). The source term reads

$$\Psi = \begin{cases} 4\eta d_e^2 & E < E_{\mathrm{pmp}} \\ 4\eta d_e^2 + \nabla \cdot (k_i \nabla T_{\mathrm{pmp}}) & E \geq E_{\mathrm{pmp}} \end{cases}, \tag{A13}$$

with $4\eta d_e^2$ representing strain heating and $(\cdot)_{\mathrm{pmp}}$ denoting quantities at the pressure melting point. For solid ice $E_{\mathrm{pmp}}$ is defined as

$$E_{\mathrm{pmp}} = E_\mathrm{s}(p) = c_\mathrm{i}(T_{\mathrm{pmp}}(p) - T_{\mathrm{ref}}) \tag{A14}$$

where $T_{\mathrm{pmp}}(p) = T_0 - \beta p$ is the pressure melting point temperature, $\beta$ is the Clausius-Clapeyron constant and $T_0$ is the melting point at standard pressure.



The temperature field $T(x,y,z)$ is then diagnostically computed from the enthalpy transfer rules

$$E(T,W,p) = \begin{cases} c_{\text{i}}(T - T_{\text{ref}}), & \text{if } E < E_{\text{pmp}} \quad \text{cold ice} \\ E_{\text{pmp}} + WL, & \text{if } E \geq E_{\text{pmp}} \quad \text{temperate ice} \end{cases} \tag{A15}$$

with $T_{\text{ref}}$ a reference temperature and $L$ the latent heat of fusion.

At the ice surface $\Gamma_s$ the boundary condition is given by the surface skin temperature with zero water content. Four different cases need to be considered for the boundary condition at the ice base $\Gamma_b$:

**Cold base (dry):** If the glacier is cold at the base and without a basal water layer (i.e. $E < E_{\text{pmp}}$ and $H_{\text{w}} = 0$), then

$$-K_{\text{c}}\nabla E \cdot \boldsymbol{n} = q_{\text{geo}}. \tag{A16}$$

**Temperate base:** If the glacier is temperate at the base without an overlying temperate ice layer with melting conditions at
        the base (i.e. $E \geq E_{\text{pmp}}$, $H_{\text{w}} > 0$ and $\nabla T' \cdot \boldsymbol{n} < \beta/K_{\text{c}}$), then

$$E = E_{\text{pmp}}. \tag{A17}$$

**Temperate ice at base:** If the glacier is temperate at the base with an overlying temperate ice layer (i.e. $E \geq E_{\text{pmp}}$, $H_{\text{w}} > 0$
        and $\nabla T' \cdot n \geq \beta/K_{\text{c}}$), we let

$$-K_0\nabla E \cdot \boldsymbol{n} = 0. \tag{A18}$$

**Cold base (wet):** If the glacier is cold but has a liquid water layer at the base which is refreezing (i.e. $E < E_{\text{pmp}}$ and $H_{\text{w}} > 0$),
        then

$$E = E_{\text{pmp}}. \tag{A19}$$

Here $T'(p) = T - T_{\text{pmp}}(p) + T_0 = T + \beta p$ is the temperature relative to the melting point, $H_{\text{w}}$ is the basal water layer thickness.
In addition to the temperate base condition, $E \geq E_{\text{pmp}}$, it is necessary to check if there is a temperate layer of ice above,
$\nabla T' \cdot \boldsymbol{n} \geq \beta/K_{\text{c}}$. The type of basal boundary condition - Neumann or Dirichlet - is therefore time dependent and therefore also
requires a nonlinear iteration.

The jump condition on $\Gamma_b$ gives the basal mass balance BMB

$$\text{BMB} = \frac{F_{\text{b}} - (\boldsymbol{q}_{\text{i}} - \boldsymbol{q}_{\text{geo}}) \cdot \boldsymbol{n}}{L\rho_{\text{i}}} \tag{A20}$$

with the frictional heating $F_{\text{b}}$ due to basal sliding, the upward heat flux in the ice $\boldsymbol{q}_{\text{i}}$, and the heat flux $\boldsymbol{q}_{\text{geo}}$ entering the ice at
the base.





### A1.5 Mass transport

Ice thickness evolution equation reads as

$$\frac{\partial H}{\partial t} = -\frac{\partial}{\partial x}\int\limits_{h_s}^{h_b} v_x dz - \frac{\partial}{\partial y}\int\limits_{h_s}^{h_b} v_y dz + \mathrm{SMB} - \mathrm{BMB} \tag{A21}$$

with $H$ the ice thickness, SMB the surface accumulation rate (surface mass balance).

### A1.6 Grounding line evolution

For HO the grounding line position is obtained from hydrostatic equilibrium: let the thickness of flotation be given by $H_f$ then

$$H_f = -\frac{\rho_{\mathrm{oc}}}{\rho_i}h_b \tag{A22}$$

with $\rho_{\mathrm{oc}}$ the density of the ocean and grounded and floating parts are defined as

$$
\begin{array}{lll}
\text{grounded} & \forall & H > H_f \\
\text{grounding line} & \forall & H = H_f \\
\text{floating} & \forall & H < H_f
\end{array}
$$


### A1.7 Evolution of the horizontal margins

The terminus evolution (for both, marine terminating glaciers, as well as ice shelves) is given by the kinematic calving front condition using a level-set method. The level set function, $\varphi$, is defined as:

$$
\begin{cases}
\varphi(\mathbf{x},t) < 0 & \mathbf{x} \in \Omega_i(t) \\
\varphi(\mathbf{x},t) = 0 & \mathbf{x} \in \Gamma_{\mathrm{cf}}(t) \\
\varphi(\mathbf{x},t) > 0 & \mathbf{x} \in \Omega_{\mathrm{c}}(t)
\end{cases}
$$

where $\Omega_{\mathrm{c}}$ is the subdomain where there is no ice. The level set therefore defines implicitly the terminus where $\varphi(\mathbf{x},t) = 0$. The level set is updated at each time step using the level set equation:

$$\frac{\partial \varphi}{\partial t} + \boldsymbol{v}_f \cdot \nabla_h \varphi = 0 \tag{A23}$$

where $\boldsymbol{v}_f$ is the ice front speed, taken as $\boldsymbol{v}_f = \boldsymbol{v}_h - c\,\mathbf{n}$ where c is the calving rate, $\boldsymbol{v}_h = (v_x, v_y)^T$ and $\nabla_h = (\partial/\partial_x, \partial/\partial_y)^T$.

For filling the required physical variables at elements that are activated due to expansion of the ice sheet area, an extrapolation

is required. This is done by solving a 2D diffusion equation for each variable $(\cdot)$

$$\nabla_h \cdot \nabla_h(\cdot) = 0 \tag{A24}$$

and assuming a diffusion coefficient of 1. Equation A24 is solved for 3D fields individually for each vertical layer $(v_x, v_y, v_z, E)$ and for $H$.





### A1.8 Surface mass balance

While in general the derivation of SMB may require an energy balance model to compute from precipitation, air temperature and radiation the surface skin temperature and SMB, we restrict use in this study a simple approach and compute surface melting is parameterized by a positive degree day (PDD) method (Reeh, 1991; Calov and Greve, 2005).

### A1.9 Convergence criteria

ISSM employs three different convergence criteria (Larour et al., 2012). A residual convergence criterion $\epsilon_{\text{res}} < \epsilon_1$ is based
on an Euclidean norm between the current and the previous iteration steps ensures the convergence of the linear system. The relative norm $\epsilon_{\text{rel}} < \epsilon_2$ uses the infinity norm to ensure convergence in relative terms. In addition, an absolute norm $\epsilon_{\text{abs}} < \epsilon_3$ controls the convergence in absolute terms. These convergence criteria are employed for the enthalpy and velocity.

### A1.10 Parameters

**Table A1.** Physical parameters used for ISSM.

| Quantity | Value |
|---|---|
| Density of ice, $\rho_{\text{i}}$ | $910\,\text{kg}\,\text{m}^{-3}$ |
| Density of the ocean, $\rho_{\text{oc}}$ | $1028\,\text{kg}\,\text{m}^{-3}$ |
| Gravitational acceleration, $g$ | $9.81\,\text{m}\,\text{s}^{-2}$ |
| Length of year, $1\,\text{a}$ | $31\,556\,926\,\text{s}$ |
| Power law exponent, $n$ | 3 |
| Flow enhancement factor, $E$ | 3 |
| Melting temperature | |
| at low pressure, $T_0$ | $273.16\,\text{K}$ |
| Reference temperature, $T_{\text{ref}}$ | $223.15\,\text{K}$ |
| Clausius-Clapeyron gradient, $\beta$ | $8.7 \times 10^{-4}\,\text{K}\,\text{m}^{-1}$ |
| Universal gas constant, $R$ | $8.314\,\text{J}\,\text{mol}^{-1}\text{K}^{-1}$ |
| Heat conductivity of ice, $\kappa$ | $2.1\,\text{W}\,\text{m}^{-1}\text{K}^{-1}$ |
| Temperate ice conductivity, $K_0$ | $0.021\,\text{kg}\,\text{m}^{-1}\text{s}^{-1}$ |
| Specific heat of ice, $c$ | $2009\,\text{J}\,\text{kg}^{-1}\text{K}^{-1}$ |
| Latent heat of ice, $L$ | $3.35 \times 10^5\,\text{J}\,\text{kg}^{-1}$ |
| Drag coefficient, $C_{\text{b}}$ | $6.72\,\text{m}\,\text{a}^{-1}\,\text{Pa}^{-1}$ |
| Sliding exponents, $(\tilde{p}, \tilde{q})$ | $(3, 2)$ |
| Sub-melt-sliding parameter, $\gamma$ | $1°\text{C}$ |



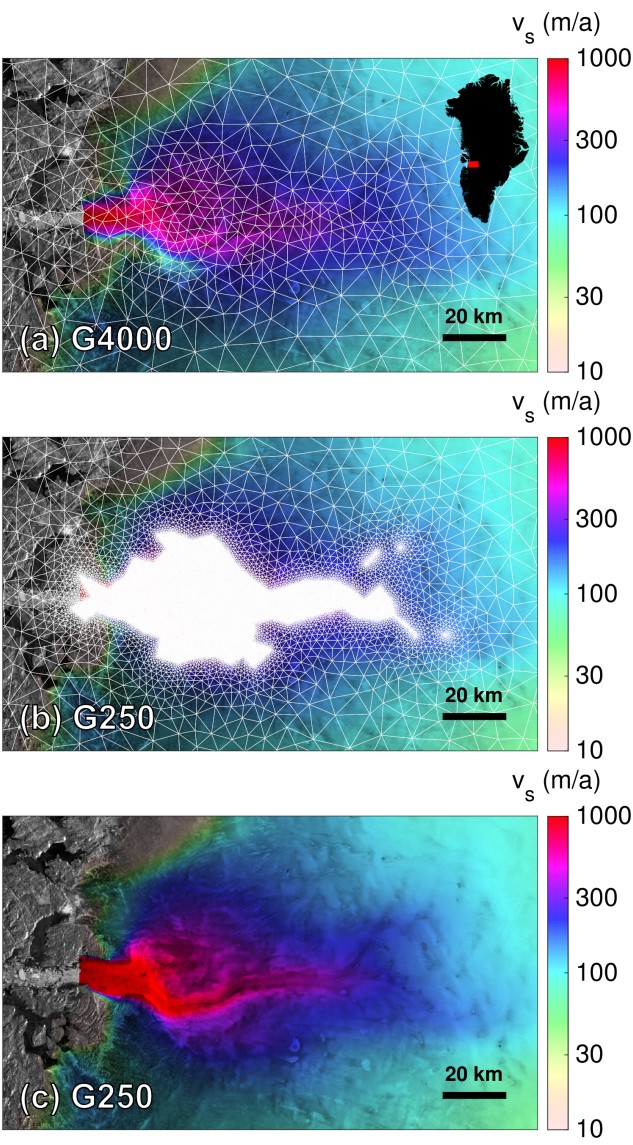

**Figure A1.** Horizontal mesh and simulated surface velocities, $v_s$, of the Jakobshavn Isbræ: (a) G4000, (b) G250, and (c) same as (b) but without mesh. The location is shown by red inset in (a). Background image is a RADARSAT Mosaic (Joughin, 2015; Joughin et al., 2016).





## A2 Mesh and horizontal velocity field

*Author contributions.* C.B., Y.F., M.R., A.H. and V.A. designed the study. Y.F. has conducted the performance measurements and instrumented the code. M.R. contributed the ISSM Greenland setup. All authors discussed the results and text. Y.F. and C.B. wrote Sect. 3 and 4. A.H. and Y.F. wrote Sect. 5, 6. A.H. wrote major parts of Sect. 1 and the Appendix. M.M. contributed ISSM design philosophy and insights into implementations. V.A. contributed the comparison to other ESM codes and strategies in terms of numerics.

*Competing interests.* The authors declare that they have no conflict of interest.

*Acknowledgements.* Calculations for this research were conducted on the Lichtenberg high-performance computer of Technical University of Darmstadt. The authors would like to thank the Hessian Competence Center for High Performance Computing – funded by the Hessian State Ministry of Higher Education, Research and the Arts – for helpful advice. The work of C.B. was partially funded by the German Science Foundation (DFG) – project number 265191195 within SFB 1194. The work of Y.F. was partially funded by the Hessian LOEWE initiative within the Software-Factory 4.0 project. A.H. thanks Simone Bnà for discussions. A.H. and M.R. acknowledge support from the German
Federal Ministry for Education and Research (BMBF) within the GROCE-2 project (grant 03F0855A), to which this work contributes advices on optimal HPC setting for coupled ice-ocean simulations.





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
