# Peer review of "A Scalability Study of the Ice-sheet and Sea-level System Model (ISSM, Version 4.18)"

_Geoscientific Model Development, 2021_

## Referee Comment (RC1)

**Review: A Scalability Study of the Ice-sheet and Sea-level System Model (ISSM, Version 4.18)**

**General Impression**

The manuscript describes the results from an instrumentation to monitor the performance of the higher-order (HO, a.k.a. Blatter-Pattyn) ice flow model including free surface and ice-front evolution as well as thermodynamics in form of an enthalpy balance, all implemented in the Ice-sheet and Sea-level System Model (ISSM). Investigations mainly focus on the scaling of different components within the modular built package. Results on scalability for parallel runs using the Message Passing Interface (MPI) standard on an Intel CPU cluster for a particular problem run on the Greenland ice-sheet are presented.

To me, topically, this manuscript fits very well into GMD. I like the detailed analysis (the authors apparently invested a lot of time) and the presentation of the scaling behaviour of different modules of the simulation workflow and the way code-profiling and timing are implemented on a, seemingly, least invasive level. The sections building the text are well structured. The figures, in general, are clear to read.

I have a few concerns with the way those scalability tests are presented and in particular how they are put in context with new architectures and exa-scale computing – I will try to elaborate in detail below and hope that this will contribute to improve the manuscript.

**Main points of critics**

I see two major points the manuscript has to be worked over before being able to be published.

**1 In parts vaguely defined scientific and computational context**

During the read of the manuscript, I came across expressions like "overall scaling" and "reasonably well" that I find difficult to put into exact context. At some places in the text I get the impression that the line of argumentation of impact of the here presented study is slightly off target, like a connection of pure MPI scalability on a limited amount of nodes to exa-scale computing. If you disagree with this, I would ask you to elaborate why MPI scalability on less than 10k cores prepares a code to be run on exa-scale machines. Also, the term "new architectures" for me is hard to interpret. Do you mean new generation CPU's or - and that what I would understand to be kind of synonymous to exa-scale computing - accelerated (GPU) systems?  Further, the title as well as the abstract gives the impression that the whole code-base of ISSM is investigated, yet, I see that investigation were focused on the HO module. I think it should come clearer that the reader should not draw conclusions to other implementations – as far as I am aware of, ISSM also offers the possibility to deploy SSA and full-Stokes. In my opinion, **you should be more precise in your definitions, and clearly state the purpose and applicability of your scalability tests**.

**2 Unexplained – and in parts unclear - settings relevant to the scalability study**

To me the circumstances and settings that lead to your scalability numbers are not completely clear. Mainly, I am missing a discussion on the influence of the compiler and flags therein (which can have a huge impact on performance) used. Partly, I am surprised by the choices made. For instance, as you ran on new generation Intel processors, to me it appears strange that you – as I understand it - did not enable AVX2, nor AVX512 (actually not even low-level vectorization)? Would you not want to optimize first the performance (by right choice of compiler flags, the least) on a single node before you test scalability on more nodes? In Byckling et al. (2017) the degenerating scalability of the matrix assembly with low polynomial order elements was clearly linked to the compiler not being able to vectorize loops. Another issue is the under-subscription of cores on the nodes (I understand that you left 50% of the cores idle) and its influence on the performance? I, frankly, do not understand the line of argumentation that this was implied by problem size. If there are enough nodes to run the problem with only half capacity, there should be also enough to double the partition amount of the equally sized problem and run it with fully subscribed nodes. Or am I missing something here, like a fixed DOFs/core ratio that needs to be maintained? Also, if you would apply platform specific flags, downclocking – or in case of undersubscription rather the lack of it – which Intel processors are known for in connection with AVX512 for thermal management – could affect the result. Additionally, could this measure have boosted performance by reducing cache-misses and thereby artificially improving scalability? Which brings me to the next question: Did you also analyse memory access and bandwidth? I think that presenting **more details on these aspects of the performance and justification of the setup would be needed** to give the reader more insight on the circumstances that lead to the presented scalability data**. Ideally,** you would redo **(one of) the tests with the full set of platform specific optimization flags** enabled **and with** – if possible – **all cores of the nodes** engaged in the computation.

**Detailed list of issues to be addressed**

The list of issues is in the order of their occurrence in the text. Quote from the manuscript are kept in blue text.

**page 1 – line 2:** It is important to test the scalability of existing ice sheet models in order to assess whether they are ready to take advantage of new cluster architectures.
As I see it, "new architectures" will be relying on parallel paradigms beyond MPI, like OpenMP threading or SIMD (e.g., Byckling et al., 2017) and on frameworks or libraries that enable the utilization of accelerators (CUDA, HIP, Sycl, Kokkos, etc.). To my understanding, the study here – as important scalability of the MPI implementation is and remains to be – does not touch these topics and I would be careful to make a direct link to exa-scale computing (which in my opinion you do in the introduction) or new architectures (or elaborate what exactly you mean by this term). This is no criticism on the here presented topic and methods, but a request to not sell a pure MPI scalability study as priming a code to be ready for new (pre)exa-scale architectures.

**page 1 – line 3:** In this paper, we discuss the overall scaling of the Ice-sheet and Sea-level System Model (ISSM) applied to the Greenland ice sheet.
What do you mean by "overall scaling"? Do you mean*: In this paper, we discuss the scaling of the MPI implementation of the HO model of the Ice-sheet and Sea-level System Model (ISSM) applied*

*to the Greenland ice sheet.* In my opinion, this statement and in particular the word "overall" can be misleading. as the study here does not allow to make conclusions to all implementations in ISSM, such as the full-Stokes model.

**page 1+2 – line 19-20:**  Today, projections are still subject to large uncertainties, which stem in particular from the climate forcing and ice-sheet model characteristics (such as the initial state, Goelzer et al., 2020).

To me this statement seems to be very much focused on sea level rise contribution from Greenland – which, admittedly, at the moment is the biggest single contributor. Yet, to my understanding (e.g., Edwards et al. 2021), the biggest uncertainty related to sea level rise is due to possible marine ice cliff instability (MICI), most prominently at the Antarctic ice-sheet, which is dominated by brittle failure and inherently difficult to be modelled by continuum models (Crawford et al., 2021). Thus, an increased resolution or throughput of continuum models does not automatically solve that particular issue – it though may help in connection with improved physics in terms of parametrizations (Crawford et al., 2021).

**page 2 – line 20-23:**  While the momentum balance can nowadays be solved using a higher-order approximation (HO, also called Blatter-Pattyn, Blatter, 1995; Pattyn, 2003) representing the physical system reasonably well, the benefits of HO pay out only if the resolution is sufficiently high, especially in the vicinity of the grounding line where the ice sheet goes afloat.

What do you mean by "reasonably well"? And in what connection? Do you mean grounding line dynamics? I would ask that you elaborate what quality of the model you are referring to and add a reference to support this statement. Does "nowadays" mean that this is a newly implemented feature in ISSM? – since I know of other HO implementations since a while.

**page 2 – line 25-28:**  As a consequence, atmosphere and ocean models will be run in the future at higher spatial resolutions, so when ice sheet codes are coupled with other components of Earth System Models (ESMs), they need to reach a level of computational performance – especially in terms of parallel scalability – comparable to that of ocean and atmosphere models.

To me, that statement implicitly renders ocean and atmosphere models to be more scalable as ice sheet models. If there is a general statement along this line to be found in literature, then please cite it here. But even if it is the case, could you shed some light to whether it is important for the combined ESM performance? From my experience, the absolute CPU consumption at least of lower order ice-sheet models falls way below that of the atmospheric and ocean components (admittedly, all depends on the resolution) – not to mention the couplers, such that improved scalability of ice-sheet models might not be that significant for the whole ESM. If you would have some figures at hand that would show the relative importance of ice-sheet models in ESM's (e.g. by relating their SYPD to other components therein), this would support your line of argumentation and improve the information to the reader.

**page 2 – line 25-28:**  To study the performance of ISSM, we select a real-life system as a test case: the Greenland Ice Sheet (GrIS) simulated at different horizontal resolutions – covering the range from what is today's standard for long-term simulations (Plach et al., 2019), such as paleo-spinups, up to the present-day highest resolutions in projections such as the ones used in the international benchmark experiment ISMIP6 (Goelzer et al., 2020; Rückamp et al., 2020a).

You study the performance of the HO-module of ISSM. To me this means that no conclusions to the performance of other approximations to Stokes equations implemented in ISSM can be drawn.

**page 2 – line 39-40:** Such an in-depth performance analysis has not yet been performed for ISSM and comes timely with the first exascale-ready ESMs (Golaz et al., 2019) ushering a new era of climate modelling.

To me this sentence suggests that scalability on MPI level would prepare a code to be ready for exa-scale computing. In my opinion this does not apply. If you disagree, please elaborate why you think this is the case.

**page 3 – line 56:** With increasing resolution as well as increasing order of shape functions, the number of DOF is increasing for a particular PDE boundary value problem.

This is a FEM specific statement referring to higher order elements and not generally might be directly understandable to every reader. Hence, I think a reference would help.

**page 3 – line 59-60:** For the software package PETSc (Balay et al., 2021a) on which ISSM is built, the recommendation is a minimum of 10 000 DOF/core.

PETSc, as I understand it, provides an interface to different available solution methods – is it really so that there is a general rule of maximum DOF/core or is this recommendation linked to a certain solution strategy of a particular problem?

**page 3 – line 76-77:** Since direct linear solvers are rather uncommon for sparse problems of such size, we rely on an iterative solver – based on Habbal et al. (2017) we selected GMRES preconditioned with block Jacobi method.

What do you mean by "uncommon"? At least in numerical glaciology MUMPS is not that uncommon (Gagliardini et al., 2013), simply because of the bad condition number of the resulting system of the Stokes problem (that is why iterative solvers demand a good pre-conditioning strategy). Secondly, can you please elaborate whether you apply GMRES only to the horizontal stress balance or to every linear solution step (enthalpy, free surface)?

**page 3 – line 81-82:** As a result, then, such an analysis can become part of the standard production environment, providing insight into the code's performance as algorithms or the code's environment change.

Can you please elaborate what you mean by "code's environment"? Do you refer to software stack, hardware, middleware?

**page 4 – line 94-95:** For this study we focus on a selected subset of the capabilities of ISSM, e.g. we employ only the HO approximation of the stress balance.

This links to previous statement that in my opinion you should drop a note on that in the abstract and introduction, too, such that readers do not link scalabilities numbers to other approximations of or the complete Stokes equation solver.

**page 4 – line 101-103:** The computation within a time step is conducted in a sequence of different modules (see Figure 1 for a schematics of the main execution substeps), which means that the different balance equations for momentum, mass and energy are not solved in a coupled fashion.

From this statement and also from Fig. 1 I understand that there is no iteration between coupled sequential modules (e.g., thermo-mechanical coupling) taking place on a single time-step. Can you clearly state that in the text and perhaps also drop a note why this is practised? And perhaps also discuss to whether this could have an influence on the stability linked to time-step size, which inherently would influence your SYPD.

**page 4 – line 110:** The results presented in this study are based on a realistic setup of the GrIS.
In what context realistic? I think that if you write that you are profiling the code using a setup that represents one that is used for projection runs, the message would be clearer.

**page 4 – line 115:** Climate forcing fields are not read in at each time step in the surface mass balance (SMB) module.
Can you please provide information on how often they are read in, as I/O might influence the performance of the code? In particular, is the climate updated during the timesteps taken for scalability tests?

**page 5 – line 119-120:** These settings basically specify a system close to an equilibrium.
This is confusing: I would assume that present day GrIS setup is anything else than close to equilibrium at its marine fronts. Can you please explain?

**page 5 – line 129-130:** Since we allow in each time step the individual modules to reach their convergence criteria, we intentionally exclude the timings from a cold-start based on a poor initial guess.
This links to my earlier question: Do you allow for iterations between by variables mutually dependent modules on a single time-step? – Or does this statement refer to non-linear iterations of single solvers, only?

**page 5 – line 135-140:** All experiments are conducted on dedicated compute nodes of the Lichtenberg HPC system with two 48-core Intel Xeon Platinum 9242 and 384 GB of main memory each, connected with an InfiniBand HDR100 network. We employ 48 MPI processes on each node pinned to NUMA nodes. More processes per node are not possible, because of memory limitations of the high resolution model G250. Each experiment runs 3 repetitions, and results fall within a standard deviation of 10%. The basis for our instrumentation is the latest ISSM public release 4.18, which is is compiled with GCC 10.2 (optimization level -O2), Open MPI 4.0.5 (Graham et al., 2006) and PETSc 3.14 (Balay et al., 2021b).
This is a very important paragraph and from my point of view needs to be elaborated in more details (see item 2 of main points). This seems to be the only place to really obtain information of the hard- and software environment and parameters used in the study. I have a few questions that I see as essential to be worked over:

1. I understand that you are compiling ISSM with gcc and you only opt for `-O2`. Checking on the cluster accessible to me, I conclude that this optimization flag excludes any vectorization and utilization of particular performance enhancing CPU features (like AVX2, or even AVX512) of that high-end CPU:
   ```
   $ gcc -O2 -E -v - </dev/null 2>&1 | grep cc1
   /path/to/gcc/10.3.0/cc1 -E -quiet -v - -mtune=generic -mtune=x86-64 -O2
   ```
   If you agree with my statement above, please explain why you on purpose drop many advantages, like vectorization, modern CPU architectures would bring along? I am just sceptic that if you would use a set of optimizations that take better use of the underlying hardware, you might get different (perhaps even worse, as communication might become a relatively stronger bottleneck) scaling. Also, if claiming to test for scalability on modern architectures and not utilizing essential built-in features of the CPUs you run on, to me is a contradiction, in particular as you mention SYPD. My suggestion would be to present runtimes and scalability obtained with code that was compiled with platform specific

optimization flags – in your case I guess this would be `-mtune=cascadelake` (see https://gcc.gnu.org/onlinedocs/gcc/x86-Options.html)

2. Are the libraries (OpenMPI and PETSc) also compiled only with simple 2nd level optimization flag?
3. Also, please explain in detail why the "memory limitations" in the G250 run imply under-subscription of cores on nodes (see my comments in "main points")
4.  Do you consistently undersubscribe and use only half of the cores in a node throughout all experiments, i.e., not just G250? If not, I would say that you then should not compare the runtimes.
5. As you base your numbers on multiple runs (which is good practise): Did you run on an empty system or did you have to share it with other users? If the latter applies, can you estimate the influence?

**page 6 – Fig. 1:**

[Figure]

I think the mass balance in the appendix is rather Eq. A9, as A10 is just the kinematic BC on the bedrock

Further, I think it would improve readability, if you would explain the numbers in the r.h.s. of the figure in the caption.

**page 8 – line 164-165:** For this it uses an even distribution of the elements, which is constant over time and independent of the modules, and each modules handles the parallelization in the same way.
Does this link to load balance through domain decomposition? Is this "even distribution" of elements obtained from within ISSM or a result of an external partitioned mesh?

**page 8 – line 166-168:** While this step is free of MPI communication, all entries which are assigned to other MPI processes are communicated in the assembly of the equation system leading to many MPI calls.
I do not completely understand this statement. Could it be that you mean while for matrix entry construction in the bulk of the partition no MPI communication is required, entries for nodal values being shared with other MPI processes on domain boundaries impose a lot of MPI traffic?

**page 8 – line 169-170:** In case of a nonlinear system, a convergence criteria has to be computed, which mainly consists of a single MPI allreduce, …
As I see it, this is a technical term that not necessarily might be understood by readers not being familiar with MPI – I would recommend at least a citation where people can look up its meaning.

**page 9 – line 216-217:** In the end, we provide Score-P with a whitelist of 52 functions to instrument and instrument by hand 6 code regions by bracketing them with Score-P instrumentation calls.
Can you please explain? Does that mean that all mentioned 52 whitelisted functions are contained within 6 distinguishable code regions?

**page 10 – line 220:** These 58 functions cover the hot paths of the main parts of the code.
I wonder if it would be better for a wider audience not familiar with code profiling to either explain the terminus or to provide a citation.

**page 10 – line 234-236:** When, in addition, the 84 billion PETSc functions are instrumented, we do not detect any noteworthy additional overhead, the same holds for the addition of the 58 million calls related to ISSM.
Looking at table 2, I would understand that this should read as "84 billion calls are made to instrumented PETSc functions" as well as "57 million calls related to ISSM" (instead of 58 million)

**page 11 – line 250:** Preallocation is not an option for the thermal module due to the physics of the system, which is why it is not preallocated.
I do not get the idea here. Can you please explain what part of the physics implies the changing sizes?

**page 11 – line 251:** Since the nonlinear iteration scheme of the thermal module only needs one to three nonlinear iterations in our setup, the reallocation does not matter as much as in the stress balance horizontal module anyway.
Focusing on the lower limit given here, I wonder how one can judge from a single non-linear iteration whether the system is converged? Or does this mean you are comparing to a previous time step? Concerning the upper bound given here: From my experience, convergence behaviour (also of the energy balance) strongly depends on the problem (and on the demanded accuracy) and I would be careful in dismissing the possibility to exceed 3 rounds. So, perhaps you mention that this situation applies to the particular problem you studied?

**page 12 – line 275-276:** Exploiting the natural anisotropy of the problem, the horizontal and vertical components are solved in an uncoupled fashion, and the structure of the PDEs varies greatly between these two components.
I understood that you investigate HO or Blatter-Pattyn, which has only horizontal components as PDEs, namely, Eqs. (A1 and A2) to solve for, simply because the vertical hydrostatic stress balance Eq. (A3) is assumed. As I see it, there is no PDE solved in vertical direction, just a quadrature is needed to deduce the vertical velocity component (A9) in a post-processing step. Thereby, I would suggest to write: *… and the structure of the solution procedure varies between these two main components.*

**page 12 – line 289-290:** While the stress balance vertical module is solved in a linear equation system, the nonlinear equation system of the stress balance horizontal module has to be solved iteratively and needs approximately twelve iterations per time step.
Like before, this is a statement I understand to only apply to the particular case investigated here and cannot be put into such a generalized form.

**page 19 – line 363-364:** So, for a fixed core count, increasing DOF increases performance but, with an increasing number of MPI processes, the communication overhead of the assembly starts dominating at some point.
That is interesting (and rather a curious question than a point of critics): Could one speculate that then a shared memory approach (OpenMP threading) on intra-node in combination with message

passing (distributed memory) only for inter-node communication could improve that situation? Would this be achievable in ISSM?

**page 20 – line 388-391:** One approach to address this problem would be to use an alternative approach for treating Dirichlet boundary conditions such as including entries for all nodes in the stiffness matrix, setting rows of constrained nodes to 0 except along the diagonal, and changing the right hand side to the value of the constraint.
Can you please explain: Why would filling up the matrix diagonal elements and the r.h.s. with the Dirichlet conditions apart from increasing the size of the system speed up computations of its solution? I can understand that scalability of this single step suffers by eliminating Dirichlet conditions, but do not completely get why the absolute runtime would be that negatively affected. What am I missing here?

**page 20 – line 392-393:** In addition, wherever a low number of DOF per MPI process is limiting scalability with increasing core counts, the number of DOF could be increased by switching to P2 (quadratic) elements or solving the full Stokes problem.
I thought that you refer to issues connected with the evolution of the lateral margin – how would switching to a full Stokes scheme have any influence on that problem? If you refer to the solution of the stress-balance, I would be careful to claim that the change from HO to (full-) Stokes will automatically increase scalability by introducing another unknown (and hence DOF), as you are buying into a more demanding saddle-point problem and most likely will have to invest into pre-conditioning strategies to solve the problem with iterative methods (e.g., Isaac et al., 2015).

**page 21 – line 413-414:** The recommendations and directions given by (Baur et al., 2021) thus will need to be adopted to the future development of ISSM too.
I my opinion it would be valid information for the reader to write out the performance criterion in the reference given and even include an idea on how far ISSM falls behind – is it orders of magnitude or just a few %? In this context I again wonder how much additional optimization flags utilizing optimized vectorization features on the CPU would improve results.

**page 21 – line 428-429:** To analyse the practical throughput of a typical application of ISSM, we conducted transient simulations for the Greenland Ice Sheet in five different horizontal resolutions.
Similar as before, I think that it would be good (in this case for readers focusing on the conclusion part, only) to elaborate "typical" by describing that you solved HO approximation.

**page 26 – line 544, Eq. (A21):** Since $h_s$ and $h_b$ (interpreting them to be the z-elevation of surface and bedrock, respectively) are entangled with the unknown variable $H = h_s - h_b$, please explain how you treat that dependence. Please, also verify in the equation that the order of integration boundaries is correct (to me they should be interchanged).

**page 27 – line 572:** These convergence criteria are employed for the enthalpy and velocity.
Can you please elaborate: Is either one of the above-mentioned criteria applied or is there some kind of mix of all three? If the latter applies, describe how this is achieved.

**List of typos and to contents less important issues**

page 1 – affiliations: Hesse →Hessen

page 2 – line 53: (e.g. triangles or quadrilaterals in  2D, tetrahedra or prisms in  3D) - remove articles

page 3 - line 79: … an Earth system scientist's view → an Earth system scientist's view

page 4 - line 104: … using a fixed-point Picard iteration whose each step involves solution of a linear equation system → ( only a suggestion): … using a fixed-point Picard iteration where each step involves the solution of a linear equation system

page 5 – line 140: missing space between sentences: … (Balay et al., 2021b).For profiling …

page 6 – line 164-165: … and each modules→module handles the parallelization in the same way.

page 8 – line 169: … a convergence criteria … → … a convergence criterion …

page 9, line 211-212 (only a suggestions): The profile contains the information whether an MPI call belongs to an assembly, the solver or some other PETSc algorithm. The "for example" reads strange to me, as to me this sentence is no specification of a general statement.

page 11 – line 250: Preallocation is not an option for the thermal module due to the physics of the system. To me the last part is redundant.

page 12- line 297: The thermal module contains a nonlinear iteration schema → scheme

page 13 – caption Fig 4: what means "(draft)" at the end of the line?

page 14 – line 302: (1) a levelset module that is computed first … ; I would say that a module is rather executed than computed

page 15 – caption Fig 5: what means "(draft)" at the end of the line? Also, why are a), b) and c) not explained in main caption but inserted as sub-captions?

page 19 – line 366: increasing run-time of mat assembly; do you mean: matrix assembly?

page 29:  there is an orphan subsection header on this page: A2    Mesh and horizontal velocity field. I presume Figure A1 of page 27 should occur under this and I simply assume that the final typesetting will correct that. Mentioning Figure A1: the white area inside JH Isbrae to me does not really convey a lot of information – I would suggest to have a zoom into the densest mesh area or else drop sub-figure b.

**References**

Byckling, M., J. Kataja, M. Klemm, and T. Zwinger (2017): *OpenMP SIMD Vectorization and Threading of the Elmer Finite Element Software*, In: de Supinski B., Olivier S., Terboven C., Chapman B., Müller M. (eds) Scaling OpenMP for Exascale Performance and Portability. IWOMP 2017. Lecture Notes in Computer Science, vol 10468. Springer, doi: 10.1007/978-3-319-65578-9_9

Crawford, A.J., D.I. Benn, J. Todd, J.A. Åström, J.N. Bassis and T. Zwinger, (2021): *Marine ice-cliff instability modeling shows mixed-mode ice-cliff failure and yields calving rate parameterization*. Nat. Commun. 12, 2701, doi:10.1038/s41467-021-23070-7

Edwards, T.L, Nowicki, S., Marzeion, B., et al. (2021): *Projected  land ice contributions to twenty-first-century sea level rise*, Nature, 593, 74–82, doi:10.1038/s41586-021-03302-y

Isaac, T., G. Stadler and O. Ghattas (2015): *Solution of Nonlinear Stokes Equations Discretized By High-Order Finite Elements on Nonconforming and Anisotropic Meshes, with Application to Ice Sheet Dynamics,* SIAM J. Sci. Comput., 37(6), B804–B833, doi: 10.1137/140974407

---

## Referee Comment (RC2)

**Review of "A Scalability Study of the Ice-sheet and Sea-level System Model (ISSM, Version 4.18) by Y. Fischler, M. Ruckamp, C. Bischof, V. Aizinger, M. Morlinghem, A. Humbert**

In the manuscript in question, the authors perform a detailed study of the overall scaling of the Ice-sheet and Sea-level System Model (ISSM) in the context of a Greenland ice sheet simulation and the higher-order (HO) Blatter-Pattyn model for the ice sheet velocities/momentum balance. The authors describe a low-overhead performance instrumentation using Score-P developed within this code base to enable continuous performance monitoring. The scalability study reveals that the matrix assembly part of the computation is the main bottleneck when it comes to scalability/performance, and should be examined further.

The manuscript in question is well-written, interesting and a good fit for GMD. My recommendation is publication following a minor revision. I ask that the authors please address the following questions/comments in their revision.

- The authors mention exascale readiness in the introduction, but there is no discussion in the paper of whether ISSM is portable to up-and-comping heterogenous architectures (GPUs). Is it? Can the present study be repeated on a set of GPUs? Some discussion of this is warranted.
- On line 77 of the introduction, the authors mention that they are using a GMRES linear solver preconditioned with a simple block Jacobi preconditioner. The HO Stokes equations are symmetric. Have the authors tried using Conjugate Gradient? I additionally worry that the Jacobi preconditioner is inadequate for problems with floating ice, e.g., Antarctica, as shown in the references by Tezaur et al. and additionally: (1) T. Isaac, G. Stadler, and O. Ghattas, Solution of nonlinear Stokes equations discretized by high-order finite elements on nonconforming and anisotropic meshes, with application to ice sheet dynamics, SIAM J. Sci. Comput., 37 (2015), pp. B804–B833, doi:10.1137/140974407 and (2) R. Tuminaro, M. Perego, I. Tezaur, A. Salinger, and S. Price. A Matrix Dependent/Algebraic Multigrid Approach for Extruded Meshes with Applications to Ice Sheet Modeling. SIAM Journal on Scientific Computing 2016 38:5, C504-C532. Are more sophisticated preconditioners required for Antarctica? Is scaling of those comparable to what you show for Greenland?
- Line 131-132: the linear solver convergence tolerances stated here seem loose to me. Have the authors verified that the solutions they have obtained at all their mesh resolutions are sufficiently converged/accurate? Accuracy/verification is an important thing to establish prior to studying scalability/performance.
- It is really great that you have set up a workflow and are doing performance monitoring! I agree that without this, it is inevitable that performance will be compromised in a big code with a lot of moving parts. It isn't entirely clear to me when the performance monitoring tests are run. Does it happen every time there is a PR merged into ISSM? Does it happen automatically or it must be run manually? Some further discussion of this is warranted. All I found was the following phrase: "it is quite feasible to periodically run an instrumented version of the code as part of the regular work of domain scientists", which suggests the performance testing is not run regularly or automatically, but perhaps I am misunderstanding.
- I did not really find discussion of how the load-balancing is done of the mesh on which the simulation proceeds, and how the mesh gets updated in a time-dependent simulation modeling

ice sheet evolution. How do you partition the mesh? Do you repartition every time the geometry (active mesh) changes? Or you partition a mesh including active and inactive cells once in the beginning? The latter approach has the potential of giving a lot of procs with no elements or poorly load-balanced meshes. Some discussion of this is warranted. I think the load imbalances you talk about in the paper have a different cause, unless I am misunderstanding.

- Is there any hope to improve the scalability of the matrix assembly?
- It would be interesting to compare ISSM performance to that of other open-source ice-sheet models based on the HO Stokes equations. I am not suggesting to do this in the paper, just commenting.

Minor comments:

- Change "to solve" in the first line of the abstract to "solving".
- Line 12: remove commas around "thus".
- Line 25: I don't really understand the phrase, "standalone ice sheet projections are suffering from a large spread in climate forcing fields". I don't think "suffering" is the right word here. Please rephrase.
- After a colon, one does not use a capital letter. In many such instances, the colon should be replaced with a period.
- Lines 131-132: I suggest stating what is \epsilon_i here so the reader does not have to refer to the appendix.
- Line 140: there is a space missing before "For profiling and tracing".
- Line 147: replace colon with period.
- Line 171: "data are stored" instead of "data is stored". Change "scalars, both" to "scalars. Both".
- Line 182: replace colon with period.
- Line 190: change "The instrumented" to "the instrumented".
- Line 193: change "Sampling" to "sampling".
- All strong scaling figures: please move the linear scaling line to be either below or above all the other curves. It's very hard to see it with all the lines on top of it (e.g., in Fig. 3).
- Line 288: change "It" to "it".
- Line 293: change "12.000" to "12 000".
- Some of the figure captions say "(draft)". Was that intentional? I suspect it was not.
- Line 420: change "to very a modest" to "to a very modest".

---

## Author Comment (AC1)

**GMD Reviews and Authors' Response concerning the paper "A Scalability Study of the Ice-sheet and Sea-level System Model (ISSM, Version 4.18)"**

Yannic Fischler[1], Martin Rückamp[2], Christian Bischof[1], Vadym Aizinger[3], Mathieu Morlighem[4,5], and Angelika Humbert[2,6]

[1]Department of Computer Science, Technical University Darmstadt, Darmstadt, Hesse, Germany
[2]Alfred-Wegener-Institut, Helmholtz-Zentrum für Polar- und Meeresforschung, Bremerhaven, Bremen, Germany
[3]Chair of Scientific Computing, University of Bayreuth, Bayreuth, Bavaria, Germany
[4]Department of Earth Sciences, Dartmouth College, Hanover, United States of America
[5]Department of Earth System Science, University of California Irvine, United States of America
[6]Faculty of Geosciences, University of Bremen, Bremen, Germany

**Correspondence:** Yannic Fischler (yannic.fischler@tu-darmstadt.de)

*Copyright statement.* ©2021 all rights reserved

**1 General Comments**

We thank Thomas Zwinger for the detailed and insightful comments. They helped us to sharpen some important issues and avoid potential misunderstandings.

In the following two sections we address each referee's comments in detail. The text in black is a verbatim rendition of the referee's comments. Our responses are typeset in the color of this paragraph underneath the referee's comments that they pertain to. In summary, the following points have been added/changed in the revised manuscript.

- The reviewers mentioned the somewhat vague discussion in the original manuscript regarding the future computing architectures and exascale computing. We clarified the pertaining points and clearly stated the goals and the scope of our study: an in-depth investigation of MPI-parallel scaling performance of ISSM and its main compute kernels.

- As requested by Thomas Zwinger, we added technical details on the used test setup – including the compiler flags and node occupancy strategy.

- We added specific model throughput goals for use of ISSM within coupled ESM-runs based on the published ocean simulation throughput data for FESOM2.

- We included several relevant references as suggested by reviewers

- In the revised version, we (will) clearly state from the abstract to conclusion, that we only assessed the higher-order Blatter-Pattyn approximation part of the code. And Thomas is definitely right, from this we cannot draw any conclusions on scalability of the entire code.

- Last but not least – we corrected throughout the manuscript a number of small mistakes, sharpened vague formulations, clarified our claims – thanks to the careful reading and constructive criticism by the reviewers!

**2 Review 1 by Thomas Zwinger**

**2.1 General Impression**

The manuscript describes the results from an instrumentation to monitor the performance of the higher-order (HO, a.k.a. Blatter-Pattyn) ice flow model including free surface and ice-front evolution as well as thermodynamics in form of an enthalpy balance, all implemented in the Ice- sheet and Sea-level System Model (ISSM). Investigations mainly focus on the scaling of different components within the modular built package. Results on scalability for parallel runs using the Message Passing Interface (MPI) standard on an Intel CPU cluster for a particular problem run on the Greenland ice-sheet are presented.

To me, topically, this manuscript fits very well into GMD. I like the detailed analysis (the authors apparently invested a lot of time) and the presentation of the scaling behaviour of different modules of the simulation workflow and the way code-profiling and timing are implemented on a, seemingly, least invasive level. The sections building the text are well structured. The figures, in general, are clear to read.

I have a few concerns with the way those scalability tests are presented and in particular how they are put in context with new architectures and exa-scale computing – I will try to elaborate in detail below and hope that this will contribute to improve the manuscript.

**2.2 Main points of critics**

I see two major points the manuscript has to be worked over before being able to be published.

**2.2.1 1 In parts vaguely defined scientific and computational context**

During the read of the manuscript, I came across expressions like "overall scaling" and "reasonably well" that I find difficult to put into exact context. At some places in the text I get the impression that the line of argumentation of impact of the here presented study is slightly off target, like a connection of pure MPI scalability on a limited amount of nodes to exa-scale computing. If you disagree with this, I would ask you to elaborate why MPI scalability on less than 10k cores prepares a code to be run on exa-scale machines. Also, the term "new architectures" for me is hard to interpret. Do you mean new generation CPU's or - and that what I would understand to be kind of synonymous to exa-scale computing - accelerated (GPU) systems? Further, the title as well as the abstract gives the impression that the whole code-base of ISSM is investigated, yet, I see that

investigation were focused on the HO module. I think it should come clearer that the reader should not draw conclusions to other implementations – as far as I am aware of, ISSM also offers the possibility to deploy SSA and full-Stokes. In my opinion, **you should be more precise in your definitions, and clearly state the purpose and applicability of your scalability tests.**

The purpose of this study was to investigate MPI-Scaling of ISSM. It is our conviction that MPI will be the backbone of any exascale architecture, and if the code already exhibits scaling issues at the number of cores that we can bring to bear in our configuration, it will be a show-stopper on the road to exascale. How exascale-systems will look like at the node level is hard to predict indeed. Accelerators and GPU-accelerated systems are certainly very good candidates, but the FUGAKU system at Riken, which employs Fujitsu's ARM-based CPU with vector extensions, claims (single-precision) exascale capabilities without GPUs (see https://www.r-ccs.riken.jp/en/fugaku/about/). Indeed, not all code parts that underly our modeling configuration are part of other models, but we believe that our scaling study sheds light on some issues that might also be of relevance for other models. We have made these points clearer in the manuscript.

**2.2.2 2 Unexplained – and in parts unclear - settings relevant to the scalability study**

To me the circumstances and settings that lead to your scalability numbers are not completely clear. Mainly, I am missing a discussion on the influence of the compiler and flags therein (which can have a huge impact on performance) used. Partly, I am surprised by the choices made. For instance, as you ran on new generation Intel processors, to me it appears strange that you – as I understand it - did not enable AVX2, nor AVX512 (actually not even low-level vectorization)? Would you not want to optimize first the performance (by right choice of compiler flags, the least) on a single node before you test scalability on more nodes? In Byckling et al. (2017) the degenerating scalability of the matrix assembly with low polynomial order elements was clearly linked to the compiler not being able to vectorize loops. Another issue is the under-subscription of cores on the nodes (I understand that you left 50% of the cores idle) and its influence on the performance? I, frankly, do not understand the line of argumentation that this was implied by problem size. If there are enough nodes to run the problem with only half capacity, there should be also enough to double the partition amount of the equally sized problem and run it with fully subscribed nodes. Or am I missing something here, like a fixed DOFs/core ratio that needs to be maintained? Also, if you would apply platform specific flags, downclocking – or in case of undersubscription rather the lack of it – which Intel processors are known for in connection with AVX512 for thermal management – could affect the result. Additionally, could this measure have boosted performance by reducing cache-misses and thereby artificially improving scalability? Which brings me to the next question: Did you also analyse memory access and bandwidth? I think that presenting **more details on these aspects of the performance and justification of the setup would be needed** to give the reader more insight on the circumstances that lead to the presented scalability data. **Ideally**, you would redo **(one of) the tests with the full set of platform specific optimization flags** enabled **and with** – if possible – **all cores of the nodes** engaged in the computation.

The issue of compiler flags and MPI-process loading of hardware cores is certainly of relevance. As we elaborate later in answering your detailed comments, more aggressive optimization flags had less than 2% impact on the overall runtime, and fully populating all CPU cores with MPI processes was not possible for the G250 model due to memory limits. The issues you raised here pertain mainly to node-level optimization, which is an important facet of performance engineering, and in a

production framework on a particular architecture these extra percent performance should be realized. For our study, however, they do not have, in our view, a significant impact, and we opted for the safe optimization levels that sidestep some of the numerical issues that could arise with more aggressive compilation. ISSM is a highly memory-bound code, and we believe that this issue has to be addressed at the algorithmic level before techniques such as vectorization can have a significant impact on overall runtime.

**2.3 Detailed list of issues to be addressed**

The list of issues is in the order of their occurrence in the text. Quote from the manuscript are kept in blue text.

**2.3.1 page 1 – line 2:**

It is important to test the scalability of existing ice sheet models in order to assess whether they are ready to take advantage of new cluster architectures.

As I see it, "new architectures" will be relying on parallel paradigms beyond MPI, like OpenMP threading or SIMD (e.g., Byckling et al., 2017) and on frameworks or libraries that enable the utilization of accelerators (CUDA, HIP, Sycl, Kokkos, etc.). To my understanding, the study here – as important scalability of the MPI implementation is and remains to be – does not touch these topics and I would be careful to make a direct link to exa-scale computing (which in my opinion you do in the introduction) or new architectures (or elaborate what exactly you mean by this term). This is no criticism on the here presented topic and methods, but a request to not sell a pure MPI scalability study as priming a code to be ready for new (pre)exa-scale architectures.

As said above, we believe that MPI will remain the main scaling vehicle. Node performance improvements with OpenMP threading and/or SIMD vector extensions will play a major role, but we believe that they will not remedy some of the issues we identify. The last paragraph in the abstract tries to clarify this issue.

**2.3.2 page 1 – line 3:**

In this paper, we discuss the overall scaling of the Ice-sheet and Sea-level System Model (ISSM) applied to the Greenland ice sheet.

What do you mean by "overall scaling"? Do you mean: In this paper, we discuss the scaling of the MPI implementation of the HO model of the Ice-sheet and Sea-level System Model (ISSM) applied to the Greenland ice sheet. In my opinion, this statement and in particular the word "overall" can be misleading. as the study here does not allow to make conclusions to all implementations in ISSM, such as the full-Stokes model.

Our intention was to discuss the scaling of all modules used in a transient simulation in addition to the scaling of individual modules. Probably 'overall' is not a good terminology for this, as we indeed do not analyse the fullStokes performance. We will rephrase the text accordingly and emphasize once more that we only analyse the performance of the HO-BP momentum balance.

**2.3.3 page 1+2 – line 19-20:**

Today, projections are still subject to large uncertainties, which stem in particular from the climate forcing and ice-sheet model characteristics (such as the initial state, Goelzer et al., 2020).

To me this statement seems to be very much focused on sea level rise contribution from Greenland – which, admittedly, at the moment is the biggest single contributor. Yet, to my understanding (e.g., Edwards et al. 2021), the biggest uncertainty related to sea level rise is due to possible marine ice cliff instability (MICI), most prominently at the Antarctic ice-sheet, which is dominated by brittle failure and inherently difficult to be modelled by continuum models (Crawford et al., 2021). Thus, an increased resolution or throughput of continuum models does not automatically solve that particular issue – it though may help in connection with improved physics in terms of parametrizations (Crawford et al., 2021).

Indeed, the initial state is only one of the ice-sheet model characteristics that is contributing to the uncertainty of projections. This sentence introduces to the broad audience that some uncertainties from the forcings and others from the model characteristics. We fully agree that resolution alone will not make poorly represented processes better represented. However, resolution was shown to have an influence to sea level projections, too - see Rückamp et al., 2020. In the revised version we cite both mentioned references.

**2.3.4 page 2 – line 20-23:**

While the momentum balance can nowadays be solved using a higher-order approximation (HO, also called Blatter-Pattyn, Blatter, 1995; Pattyn, 2003) representing the physical system reasonably well, the benefits of HO pay out only if the resolution is sufficiently high, especially in the vicinity of the grounding line where the ice sheet goes afloat.

What do you mean by "reasonably well"? And in what connection? Do you mean grounding line dynamics? I would ask that you elaborate what quality of the model you are referring to and add a reference to support this statement. Does "nowadays" mean that this is a newly implemented feature in ISSM? – since I know of other HO implementations since a while.

Many thanks for pointing this out. We intended to say that HO is representing the physical system reasonably well, but only if the resolution is sufficiently high. Our sentence was confusing, indeed. We have added a reference for a comparison at grounding lines (MISMIP+), but the only other study for a comparison of full Stokes and HO (that kept the implementation and discretisation the same between both) we are aware of, is a manuscript currently under review in TCD (tc-2021-193), which we cannot cite right now.

**2.3.5 page 2 – line 25-28:**

As a consequence, atmosphere and ocean models will be run in the future at higher spatial resolutions, so when ice sheet codes are coupled with other components of Earth System Models (ESMs), they need to reach a level of computational performance – especially in terms of parallel scalability – comparable to that of ocean and atmosphere models.

To me, that statement implicitly renders ocean and atmosphere models to be more scalable as ice sheet models. If there is a general statement along this line to be found in literature, then please cite it here. But even if it is the case, could you shed some

light to whether it is important for the combined ESM performance? From my experience, the absolute CPU consumption at least of lower order ice-sheet models falls way below that of the atmospheric and ocean components (admittedly, all depends on the resolution) – not to mention the couplers, such that improved scalability of ice-sheet models might not be that significant for the whole ESM. If you would have some figures at hand that would show the relative importance of ice-sheet models in ESM's (e.g. by relating their SYPD to other components therein), this would support your line of argumentation and improve the information to the reader.

We certainly have no reasons to claim that ocean or atmosphere models scale any better than ice sheet models. The point we were trying to make is pretty obvious – in order to achieve a reasonable scaling for a prospective ESM, each component needs to scale well. This has been pointed out by Maisonnave, E., 2017: IS-ENES2 HighRes ESM performance resulting from OASIS updates Technical Report, TR/CMGC/17/14, CECI, UMR CERFACS/CNRS No5318, France: 'The increasing parallelism of climate models, particularly for high resolution configurations as the ones used in this work-package, necessarily requires scalability for each of their components (Amdahl's law).' Of course, in the current situation the bulk of CPU-resources is consumed by expensive ocean and atmosphere models (as well as by other ESM components), therefore the expenses and the scalability of an ice-sheet model may be of secondary importance. However, as the other ESM components are continuously evolving so has the ice-sheet model too – otherwise, the chances are that it may become a bottleneck in a future ESM.

**2.3.6 page 2 – line 25-28:**

To study the performance of ISSM, we select a real-life system as a test case: the Greenland Ice Sheet (GrIS) simulated at different horizontal resolutions – covering the range from what is today's standard for long-term simulations (Plach et al., 2019), such as paleo-spinups, up to the present-day highest resolutions in projections such as the ones used in the international benchmark experiment ISMIP6 (Goelzer et al., 2020; Rückamp et al., 2020a).

You study the performance of the HO-module of ISSM. To me this means that no conclusions to the performance of other approximations to Stokes equations implemented in ISSM can be drawn.

This is true. We will make sure in the revised version that this is clearly stated.

**2.3.7 page 2 – line 39-40:**

Such an in-depth performance analysis has not yet been performed for ISSM and comes timely with the first exascale-ready ESMs (Golaz et al., 2019) ushering a new era of climate modelling.

To me this sentence suggests that scalability on MPI level would prepare a code to be ready for exa-scale computing. In my opinion this does not apply. If you disagree, please elaborate why you think this is the case.

MPI remains the main programming model/API for large-scale systems. To illustrate, the 24 applications chosen for the Exascale Computing Project (ECP, see https://www.exascaleproject.org/research/#application) all rely on MPI, with other and diverse technologies for exploiting shared-memory and accelerator features at the node level. So MPI scalability is necessary, but not sufficient, for reaching exascale. A sentence to that effect has been added.

**2.3.8    page 3 – line 56:**

With increasing resolution as well as increasing order of shape functions, the number of DOF is increasing for a particular PDE boundary value problem.

This is a FEM specific statement referring to higher order elements and not generally might be directly understandable to every reader. Hence, I think a reference would help.

This is a good point and will indeed help readers. We will cite Wriggers 'Nonlinear Finite Element Methods' in the revised version.

**2.3.9    page 3 – line 59-60:**

For the software package PETSc (Balay et al., 2021a) on which ISSM is built, the recommendation is a minimum of 10 000 DOF/core.

PETSc, as I understand it, provides an interface to different available solution methods – is it really so that there is a general rule of maximum DOF/core or is this recommendation linked to a certain solution strategy of a particular problem?

Indeed, PETSc provides different solution methods, but for solving linear matrices, there is a recommendation for the MINIMUM (not the maximum) DOF/core. The background for this is, that with lower numbers of DOF, the cost of communication becomes too high compared to local work. However, beside the recommendation on their website, we did not find any scientific paper discussing this issue further.

**2.3.10    page 3 – line 76-77:**

Since direct linear solvers are rather uncommon for sparse problems of such size, we rely on an iterative solver – based on Habbal et al. (2017) we selected GMRES preconditioned with block Jacobi method.

What do you mean by "uncommon"? At least in numerical glaciology MUMPS is not that uncommon (Gagliardini et al., 2013), simply because of the bad condition number of the resulting system of the Stokes problem (that is why iterative solvers demand a good pre-conditioning strategy). Secondly, can you please elaborate whether you apply GMRES only to the horizontal stress balance or to every linear solution step (enthalpy, free surface)?

This is indeed not written well. We intended to address that the size of the problem calls for an iterative solver and basically only want to give a rational for using this particular iterative solver. We will rephrase this accordingly.

**2.3.11    page 3 – line 81-82:**

As a result, then, such an analysis can become part of the standard production environment, providing insight into the code's performance as algorithms or the code's environment change.

Can you please elaborate what you mean by "code's environment"? Do you refer to software stack, hardware, middleware?

Examples for what "environment" may mean, as suggested by you, have been added for clarification.

**2.3.12 page 4 – line 94-95:**

205 For this study we focus on a selected subset of the capabilities of ISSM, e.g. we employ only the HO approximation of the stress balance.

This links to previous statement that in my opinion you should drop a note on that in the abstract and introduction, too, such that readers do not link scalabilities numbers to other approximations of or the complete Stokes equation solver.

Yes, we agree and make sure that this is clear in abstract and introduction already.

**2.3.13 page 4 – line 101-103:**

The computation within a time step is conducted in a sequence of different modules (see Figure 1 for a schematics of the main execution substeps), which means that the different balance equations for momentum, mass and energy are not solved in a coupled fashion.

From this statement and also from Fig. 1 I understand that there is no iteration between coupled sequential modules (e.g., thermo-mechanical coupling) taking place on a single time-step. Can you clearly state that in the text and perhaps also drop a note why this is practised? And perhaps also discuss to whether this could have an influence on the stability linked to time-step size, which inherently would influence your SYPD.

The enthalpy balance is solved first and subsequently the stress module is solved (that includes the momentum and mass balances and the rheology). There is no iteration between these two modules. In the past, this procedure has been shown to be sufficient (Seroussi, H., Morlighem, M., Rignot, E., Khazendar, A., Larour, E., & Mouginot, J. (2013). Dependence of century-scale projections of the Greenland ice sheet on its thermal regime. Journal of Glaciology, 59(218), 1024-1034. doi:10.3189/2013JoG13J054). We will state this clearly in the revised version. We are not aware of any study that has investigated if there is an influence of this on stability or on number of linear iterations.

**2.3.14 page 4 – line 110:**

225 The results presented in this study are based on a realistic setup of the GrIS.

In what context realistic? I think that if you write that you are profiling the code using a setup that represents one that is used for projection runs, the message would be clearer.

The is indeed correct. Many thanks for this comment!

**2.3.15 page 4 – line 115:**

230 Climate forcing fields are not read in at each time step in the surface mass balance (SMB) module.

Can you please provide information on how often they are read in, as I/O might influence the performance of the code? In particular, is the climate updated during the timesteps taken for scalability tests?

This was written slightly confusing, the PDD module is executed at each time step. The module reads the climate data for the whole year and compute the yearly averaged surface mass balance. However, we intended to mention that no update of the climate fields is performed, but we edited it now to make it clearer.

**2.3.16 page 5 – line 119-120:**

These settings basically specify a system close to an equilibrium.

This is confusing: I would assume that present day GrIS setup is anything else than close to equilibrium at its marine fronts. Can you please explain?

We do not think that this is confusing, as it refers to the sentence prior to it, where we state that the calving rate equals the ice velocity, which would be an equilibrium.

**2.3.17 page 5 – line 129-130:**

Since we allow in each time step the individual modules to reach their convergence criteria, we intentionally exclude the timings from a cold-start based on a poor initial guess.

This links to my earlier question: Do you allow for iterations between by variables mutually dependent modules on a single time-step? – Or does this statement refer to non-linear iterations of single solvers, only?

We do not allow iteration between temperature and velocity, but we do allow iteration between viscosity and velocity. This is actually clearly shown in Figure 1. As mentioned before, we will include another sentence to avoid confusion.

**2.3.18 page 5 – line 135-140:**

All experiments are conducted on dedicated compute nodes of the Lichtenberg HPC system with two 48-core Intel Xeon Platinum 9242 and 384 GB of main memory each, connected with an InfiniBand HDR100 network. We employ 48 MPI processes on each node pinned to NUMA nodes. More processes per node are not possible, because of memory limitations of the high resolution model G250. Each experiment runs 3 repetitions, and results fall within a standard deviation of 10%. The basis for our instrumentation is the latest ISSM public release 4.18, which is is compiled with GCC 10.2 (optimization level -O2), Open MPI 4.0.5 (Graham et al., 2006) and PETSc 3.14 (Balay et al., 2021b).

This is a very important paragraph and from my point of view needs to be elaborated in more details (see item 2 of main points). This seems to be the only place to really obtain information of the hard- and software environment and parameters used in the study. I have a few questions that I see as essential to be worked over:

1. I understand that you are compiling ISSM with gcc and you only opt for -O2. Checking on the cluster accessible to me, I conclude that this optimization flag excludes any vectorization and utilization of particular performance enhancing CPU features (like AVX2, or even AVX512) of that high-end CPU:

   $ gcc -O2 -E -v - </dev/null 2>&1 | grep cc1

   /path/to/gcc/1 0.3.0/cc1 -E -quiet -v - -mtune=generic -mtune=x86-64 -O2

If you agree with my statement above, please explain why you on purpose drop many advantages, like vectorization, modern CPU architectures would bring along? I am just sceptic that if you would use a set of optimizations that take better use of the underlying hardware, you might get different (perhaps even worse, as communication might become a relatively stronger bottleneck) scaling. Also, if claiming to test for scalability on modern architectures and not utilizing essential built-in features of the CPUs you run on, to me is a contradiction, in particular as you mention SYPD. My suggestion would be to present runtimes and scalability obtained with code that was compiled with platform specific optimization flags – in your case I guess this would be -mtune=cascadelake (see https://gcc.gnu.org/onlinedocs/gcc/x86-Options.html)

This is a good point, as overall performance is the ultimate goal of scalability, and a scalable but slow code is not what we want. Our decision is based on the following facts:

- We compiled both ISSM and PETSc with "-O2" as well as with "-O3 -march=cascadelake -mtune=cascadelake". On a 480-core configuration, the entire calculation (without loading the model), took 1955 seconds compiled with -O2, and 1930 seconds when compiled with higher-level optimization. With 1536 cores, we observed execution times of 763 and 748 seconds, respectively. So the compiler optimization level has an impact of less than 2% in both cases. ISSM is massively memory-bound.

- As the impact of the more aggressive compiler optimizations is low, we stick to -O2 as it avoids some numerical issues that can arise with more aggressive compiler options.

- You mentioned that PETSc scales better with higher optimization level. We have observed that the scaling issues are not located in PETSc itself but arise within the MPI calls. Specifically, we observed that the assembly of the extrapolation module, which is one of the global scaling issues, spends about 75% of its runtime in the MPI library.

2. Are the libraries (OpenMPI and PETSc) also compiled only with simple 2nd level optimization flag?

   OpenMPI and PETSc are part of the module tree provided for the Lichtenberg HPC system by the computing center and these versions are also compiled with 2nd level optimization.

3. Also, please explain in detail why the "memory limitations" in the G250 run imply under- subscription of cores on nodes (see my comments in "main points")

   The Lichtenberg HPC system has 384 GB main memory per node, each of which has two 48-core CPUs. As a result, when a MPI-process runs on every one of the 96 cores, it has at most 4 GB available. However, at the G250 resolution, each MPI process requires 6.2 GB of memory.

4. Do you consistently undersubscribe and use only half of the cores in a node throughout all experiments, i.e., not just G250? If not, I would say that you then should not compare the runtimes.

   Yes, we use only half of the cores for each resolution of the model to keep the timings comparable.

295     5. As you base your numbers on multiple runs (which is good practise): Did you run on an empty system or did you have to share it with other users? If the latter applies, can you estimate the influence?

During our experiments each individual node is only used by our job. The entire cluster employs a HDR-100 Network with point-to-point connections for all nodes, see https://www.hhlr.tu-darmstadt.de/hhlr/betrieb/hardware_hlr/topologie/index.de.jsp. As a result, our timings are not affected by other users' jobs that run on other nodes.

**2.3.19    page 6 – Fig. 1:**

Figure

I think the mass balance in the appendix is rather Eq. A9, as A10 is just the kinematic BC on the bedrock

Further, I think it would improve readability, if you would explain the numbers in the r.h.s. of the figure in the caption.

That is correct, A9 is the mass balance. We will change this in the revised version and also include an explanation of the numbers in the caption, that just slipped through - many thanks for pointing this out.

**2.3.20    page 8 – line 164-165:**

For this it uses an even distribution of the elements, which is constant over time and independent of the modules, and each modules handles the parallelization in the same way.

Does this link to load balance through domain decomposition? Is this "even distribution" of elements obtained from within ISSM or a result of an external partitioned mesh?

The distribution of the elements is computed by ParMETIS within the ISSM. ISSM weights every element equally, which leads to an even distribution. In terms of load balancing this is not optimal within the stress balance module, but trying to weight the elements to achieve an optimal load balancing within the stress balance module effects other modules negatively. While the scaling of computation of entries of the equation system, which is the only part we can report as being load imbalanced, is close to be linear, the major problem of the scaling of the ISSM is the assembly of the equation system and achieving optimal load balancing of all modules is challenging, we tend to ignore load balancing at this point.

**2.3.21    page 8 – line 166-168:**

While this step is free of MPI communication, all entries which are assigned to other MPI processes are communicated in the assembly of the equation system leading to many MPI calls.

I do not completely understand this statement. Could it be that you mean while for matrix entry construction in the bulk of the partition no MPI communication is required, entries for nodal values being shared with other MPI processes on domain boundaries impose a lot of MPI traffic?

The creation of the entries of the equation system is free of communication. The entire communication happens after the entries are computed, in the assembly. We changed to formulation to make this clearer.

**2.3.22 page 8 – line 169-170:**

In case of a nonlinear system, a convergence criteria has to be computed, which mainly consists of a single MPI allreduce, ...

As I see it, this is a technical term that not necessarily might be understood by readers not being familiar with MPI – I would recommend at least a citation where people can look up its meaning.

We left out this technical detail, as it is not really necessary.

**2.3.23 page 9 – line 216-217:**

In the end, we provide Score-P with a whitelist of 52 functions to instrument and instrument by hand 6 code regions by bracketing them with Score-P instrumentation calls.

Can you please explain? Does that mean that all mentioned 52 whitelisted functions are contained within 6 distinguishable code regions?

The functionality of Score-P whitelist instrumentation is limited to the instrumentation of entire functions or methods. To gain more detailed information on the creation of the equation system, which is implemented in one function, we added explicit calls to the Score-P measurement environment for 6 code regions within this function. In the end, we covered 52 regions (functions/methods) by the whitelist mechanism and 6 code regions by hand. We clarified this in the paper.

**2.3.24 page 10 – line 220:**

These 58 functions cover the hot paths of the main parts of the code.

I wonder if it would be better for a wider audience not familiar with code profiling to either explain the terminus or to provide a citation.

Explanation was added: the modules and call paths where most of the computing time is spent.

**2.3.25 page 10 – line 234-236:**

When, in addition, the 84 billion PETSc functions are instrumented, we do not detect any noteworthy additional overhead, the same holds for the addition of the 58 million calls related to ISSM.

Looking at table 2, I would understand that this should read as "84 billion calls are made to instrumented PETSc functions" as well as "57 million calls related to ISSM" (instead of 58 million)

Yes, we have changed that, thank you for the clarification.

**2.3.26 page 11 – line 250:**

Preallocation is not an option for the thermal module due to the physics of the system, which is why it is not preallocated.

I do not get the idea here. Can you please explain what part of the physics implies the changing sizes?

Thanks for asking the right question – the reason for not using preallocation is only indirectly connected to the physics of the system; in fact, the main cause is the possibility of change in the type of the basal boundary condition for the thermal core

355 (Dirichlet <–> Neumann) that necessitates adjustments in the discretization stencil for the corresponding boundary elements. Thus the number of off-diagonal blocks may vary for different iterations. We adapted the passage in the manuscript accordingly.

**2.3.27    page 11 – line 251:**

Since the nonlinear iteration scheme of the thermal module only needs one to three nonlinear iterations in our setup, the reallocation does not matter as much as in the stress balance horizontal module anyway.

360    Focusing on the lower limit given here, I wonder how one can judge from a single non-linear iteration whether the system is converged? Or does this mean you are comparing to a previous time step? Concerning the upper bound given here: From my experience, convergence behaviour (also of the energy balance) strongly depends on the problem (and on the demanded accuracy) and I would be careful in dismissing the possibility to exceed 3 rounds. So, perhaps you mention that this situation applies to the particular problem you studied?

365    For the first iteration (the first solution) the convergence criteria are assessed with respect to the solution of the last time step. If this convergence criterion is met, we have one iteration. Three iterations is not a set number of maximum iterations, it is what we observe from the runs.

**2.3.28    page 12 – line 275-276:**

Exploiting the natural anisotropy of the problem, the horizontal and vertical components are solved in an uncoupled fashion,
370    and the structure of the PDEs varies greatly between these two components.
    I understood that you investigate HO or Blatter-Pattyn, which has only horizontal components as PDEs, namely, Eqs. (A1 and A2) to solve for, simply because the vertical hydrostatic stress balance Eq. (A3) is assumed. As I see it, there is no PDE solved in vertical direction, just a quadrature is needed to deduce the vertical velocity component (A9) in a post-processing step. Thereby, I would suggest to write: ... and the structure of the solution procedure varies between these two main components.
375    Many thanks, very good suggestion!

**2.3.29    page 12 – line 289-290:**

While the stress balance vertical module is solved in a linear equation system, the nonlinear equation system of the stress balance horizontal module has to be solved iteratively and needs approximately twelve iterations per time step.
    Like before, this is a statement I understand to only apply to the particular case investigated here
380    That is correct. We added 'for this particular application' to make sure no general conclusions on SSA or fullStokes are drawn from this.

**2.3.30    page 19 – line 363-364:**

So, for a fixed core count, increasing DOF increases performance but, with an increasing number of MPI processes, the communication overhead of the assembly starts dominating at some point.

385 That is interesting (and rather a curious question than a point of critics): Could one speculate that then a shared memory approach (OpenMP threading) on intra-node in combination with message passing (distributed memory) only for inter-node communication could improve that situation? Would this be achievable in ISSM?

We agree with this speculation. ISSM is at this point not implemented entirely in a thread-safe fashion, so some clean-up is needed before one can pursue this further. We intend, however, to look into this issue in further research.

390 ### 2.3.31 page 20 – line 388-391:

One approach to address this problem would be to use an alternative approach for treating Dirichlet boundary conditions such as including entries for all nodes in the stiffness matrix, setting rows of constrained nodes to 0 except along the diagonal, and changing the right hand side to the value of the constraint.

Can you please explain: Why would filling up the matrix diagonal elements and the r.h.s. with the Dirichlet conditions apart

395 from increasing the size of the system speed up computations of its solution? I can understand that scalability of this single step suffers by eliminating Dirichlet conditions, but do not completely get why the absolute runtime would be that negatively affected. What am I missing here?

At this point we are not entirely sure that using our suggestion would substantially improve the total runtime; the idea behind this approach is that currently the assembly of constrained degrees of freedom causes a large MPI-communication overhead

400 since the DOF elimination is done for all Dirichlet-connected nodes and involves MPI-communication for each update of the stiffness matrix. The approach with the larger matrix does not need any additional communication for the assembly part (although, one has a somewhat larger system, and the same information is then propagated during the solution procedure). However, since the matrix assembly currently represents the biggest bottleneck, avoiding this additional communication may help.

405 ### 2.3.32 page 20 – line 392-393:

In addition, wherever a low number of DOF per MPI process is limiting scalability with increasing core counts, the number of DOF could be increased by switching to P2 (quadratic) elements or solving the full Stokes problem.

I thought that you refer to issues connected with the evolution of the lateral margin – how would switching to a full Stokes scheme have any influence on that problem? If you refer to the solution of the stress-balance, I would be careful to claim

410 that the change from HO to (full-) Stokes will automatically increase scalability by introducing another unknown (and hence DOF), as you are buying into a more demanding saddle-point problem and most likely will have to invest into pre- conditioning strategies to solve the problem with iterative methods (e.g., Isaac et al., 2015).

Yes, it is correct that solving the full Stokes problem is itself not making the problem better with respect to scalability and more challenging anyway. We will skip the second part of the sentence in the revised version.

415 **2.3.33 page 21 – line 413-414:**

The recommendations and directions given by (Baur et al., 2021) thus will need to be adopted to the future development of ISSM too.

I my opinion it would be valid information for the reader to write out the performance criterion in the reference given and even include an idea on how far ISSM falls behind – is it orders of magnitude or just a few %? In this context I again wonder
420 how much additional optimization flags utilizing optimized vectorization features on the CPU would improve results.

This is an excellent point. We added a corresponding passage to the manuscript where estimated improvements in model throughput are calculated by comparing to the actual FESOM2 throughput. This specific comparison makes in our opinion more sense than using very generally formulated criteria in Bauer et al., 2021.

**2.3.34 page 21 – line 428-429:**

425 To analyse the practical throughput of a typical application of ISSM, we conducted transient simulations for the Greenland Ice Sheet in five different horizontal resolutions.

Similar as before, I think that it would be good (in this case for readers focusing on the conclusion part, only) to elaborate "typical" by describing that you solved HO approximation.

We will also here make sure that it is clear also in the conclusion that we were only concerned with HO-BP.

430 **2.3.35 page 26 – line 544, Eq. (A21):**

Since $h_s$ and $h_b$ (interpreting them to be the z-elevation of surface and bedrock, respectively) are entangled with the unknown variable $H = h_s - h_b$, explain how you treat that dependence. Please, also verify in the equation that the order of integration boundaries is correct (to me they should be interchanged).

The integration boundaries were indeed flipped. Many thanks for recognising such details - very much appreciated! The
435 bed topography is h_b. For grounded ice the surface topography is h_s=h_b+H. For floating ice h_b=-H*rho_i/rho_sw and h_s=H*(1-rho_i/rho_sw).

**2.3.36 page 27 – line 572:**

These convergence criteria are employed for the enthalpy and velocity.

Can you please elaborate: Is either one of the above-mentioned criteria applied or is there some kind of mix of all three? If
440 the latter applies, describe how this is achieved.

On page 5, line 131-133 we gave the details for the enthalpy and stress balance. However, we do agree that this sentence is confusing and we have deleted it for the revised version.

**2.4 List of typos and to contents less important issues**

– page 1 – affiliations: Hesse $\rightarrow$ Hessen

445 Hesse is how Hessen is called in English.

– page 2 – line 53: (e.g. triangles or quadrilaterals in the 2D, tetrahedra or prisms in the 3D) - remove articles

done

– page 3 - line 79: ... an Earth systems scientist's view → an Earth system scientist's view

done

450 – page 4 - line 104: ... using a fixed-point Picard iteration whose each step involves solution of a linear equation system → ( only a suggestion): ... using a fixed-point Picard iteration where each step involves the solution of a linear equation system

using a fixed-point Picard iteration whose each step involves solution of a linear equation system.

Thanks, we implemented this suggestion.

455 – page 5 – line 140: missing space between sentences: ... (Balay et al., 2021b).For profiling ...

done

– page 6 – line 164-165: ... and each modules—→module handles the parallelization in the same way.

done

– page 8 – line 169: ... a convergence criteria ... —→ ... a convergence criterion ...

460 done

– page 9, line 211-212 (only a suggestions): So the profile contains the information whether an MPI call belongs to an assembly, the solver or some other PETSc algorithm, for example. The "for example" reads strange to me, as to me this sentence is no specification of a general statement.

"for example" replaced by "respectively".

465 – page 11 – line 250: Preallocation is not an option for the thermal module due to the physics of the system, which is why it is not preallocated. To me the last part is redundant.

done

– page 12- line 297: The thermal module contains a nonlinear iteration schema → scheme

done

470 – page 13 – caption Fig 4: what means "(draft)" at the end of the line?

done

– page 14 – line 302: (1) a levelset module that is computed first ... ; I would say that a module is rather executed than computed

done

– page 15 – caption Fig 5: what means "(draft)" at the end of the line?

done

Also, why are a), b) and c) not explained in main caption but inserted as sub-captions?

– page 19 – line 366: increasing run-time of mat assembly; do you mean: matrix assembly?

done

– page 29: there is an orphan subsection header on this page: A2 Mesh and horizontal velocity field. I presume Figure A1 of page 27 should occur under this and I simply assume that the final typesetting will correct that. Mentioning Figure A1: the white area inside JH Isbrae to me does not really convey a lot of information – I would suggest to have a zoom into the densest mesh area or else drop sub-figure b.

The orphan header can only be fixed by the final typesetters. It is indeed correct, that the white area in Fig. A1 does not allow to really see much of the mesh, but when zooming out, one has not comparison with the coarse grid. We think that the figure gives an impression of the grid, without spending too many panels or subfigures on it. But if requested, we are definitely happy to include more subfigures with zooms.

**2.5 References**

– Byckling, M., J. Kataja, M. Klemm, and T. Zwinger (2017): OpenMP SIMD Vectorization and Threading of the Elmer Finite Element Software, In: de Supinski B., Olivier S., Terboven C., Chapman B., Müller M. (eds) Scaling OpenMP for Exascale Performance and Portability. IWOMP 2017. Lecture Notes in Computer Science, vol 10468. Springer, doi: 10.1007/978-3-319-65578-9_9

As discussed, we do not believe that these techniques will be profitable at the current time due to the memory-boundedness of ISSM.

– Crawford, A.J., D.I. Benn, J. Todd, J.A. Åström, J.N. Bassis and T. Zwinger, (2021): Marine ice-cliff instability modeling shows mixed-mode ice-cliff failure and yields calving rate parameterization. Nat. Commun. 12, 2701, doi:10.1038/s41467-021-23070-7

done

– Edwards, T.L, Nowicki, S., Marzeion, B., et al. (2021): Projected land ice contributions to twenty- first-century sea level rise, Nature, 593, 74–82, doi:10.1038/s41586-021-03302-y

done

– Isaac, T., G. Stadler and O. Ghattas (2015): Solution of Nonlinear Stokes Equations Discretized By High-Order Finite Elements on Nonconforming and Anisotropic Meshes, with Application to Ice Sheet Dynamics, SIAM J. Sci. Comput., 37(6), B804–B833, doi: 10.1137/140974407

505     done

---

## Author Comment (AC2)

**GMD Reviews and Authors' Response concerning the paper "A Scalability Study of the Ice-sheet and Sea-level System Model (ISSM, Version 4.18)"**

Yannic Fischler[1], Martin Rückamp[2], Christian Bischof[1], Vadym Aizinger[3], Mathieu Morlighem[4,5], and Angelika Humbert[2,6]

[1]Department of Computer Science, Technical University Darmstadt, Darmstadt, Hesse, Germany
[2]Alfred-Wegener-Institut, Helmholtz-Zentrum für Polar- und Meeresforschung, Bremerhaven, Bremen, Germany
[3]Chair of Scientific Computing, University of Bayreuth, Bayreuth, Bavaria, Germany
[4]Department of Earth Sciences, Dartmouth College, Hanover, United States of America
[5]Department of Earth System Science, University of California Irvine, United States of America
[6]Faculty of Geosciences, University of Bremen, Bremen, Germany

**Correspondence:** Yannic Fischler (yannic.fischler@tu-darmstadt.de)

*Copyright statement.* ©2021 all rights reserved

**1 General Comments**

We thank the anonymous referee for the detailed and insightful comments. They helped us to sharpen some important issues and avoid potential misunderstandings.

In the following two sections we address each referee's comments in detail. The text in black is a verbatim rendition of the referee's comments. Our responses are typeset in the color of this paragraph underneath the referee's comments that they pertain to. In summary, the following points have been added/changed in the revised manuscript.

– The reviewers mentioned the somewhat vague discussion in the original manuscript regarding the future computing architectures and exascale computing. We clarified the pertaining points and clearly stated the goals and the scope of our study: an in-depth investigation of MPI-parallel scaling performance of ISSM and its main compute kernels.

– We added specific model throughput goals for use of ISSM within coupled ESM-runs based on the published ocean simulation throughput data for FESOM2.

– We included several relevant references as suggested by reviewers

– In the revised version, we (will) clearly state from the abstract to conclusion, that we only assessed the higher-order Blatter-Pattyn approximation part of the code.

– Last but not least – we corrected throughout the manuscript a number of small mistakes, sharpened vague formulations, clarified our claims – thanks to the careful reading and constructive criticism by the reviewers!

**2   Review 2 by an anonymous referee**

In the manuscript in question, the authors perform a detailed study of the overall scaling of the Ice-sheet and Sea-level System

20   Model (ISSM) in the context of a Greenland ice sheet simulation and the higher- order (HO) Blatter-Pattyn model for the ice sheet velocities/momentum balance. The authors describe a low-overhead performance instrumentation using Score-P developed within this code base to enable continuous performance monitoring. The scalability study reveals that the matrix assembly part of the computation is the main bottleneck when it comes to scalability/performance, and should be examined further.

25   The manuscript in question is well-written, interesting and a good fit for GMD. My recommendation is publication following a minor revision. I ask that the authors please address the following questions/comments in their revision.

– The authors mention exascale readiness in the introduction, but there is no discussion in the paper of whether ISSM is portable to up-and-comping heterogenous architectures (GPUs). Is it? Can the present study be repeated on a set of GPUs? Some discussion of this is warranted.

30   We have toned down the exascale issue. The purpose of this study was to investigate MPI-Scaling of ISSM. It is our conviction that MPI will be the backbone of any exascale architecture, and if the code already exhibits scaling issues at the number of cores that we can bring to bear in our configuration, it will be a show-stopper on the road to exascale. How exascale-systems will look like at the node level is hard to predict indeed. GPU-accelerated systems are a possibility, but the FUGAKU system at Riken, which employs Fujitsu's ARM-based CPU with vector extensions,

35   claims (single-precision) exascale capabilities without GPUs (see https://www.r-ccs.riken.jp/en/fugaku/about/). The importance of MPI scaling is exemplified, for example, by the 24 applications chosen for the exascale computing project (see https://www.exascaleproject.org/, which employ MPI plus other techniques for exploiting GPUs and/or thread-level parallelism. Node-level issues, such as GPU accelerators or vector units, are an important possible issue in improving node performance, but orthogonal to this issue. For example, Brædstrup et al. (2014) presents a GPU implementation

40   of an ice-sheet model. But they does not run all modules ISSM runs. We noted that ISSM is massively memory bound which suggests that significant algorithmic work needs to be invested before ISSM can profitably use GPUs or vector acceleration.

– On line 77 of the introduction, the authors mention that they are using a GMRES linear solver preconditioned with a simple block Jacobi preconditioner. The HO Stokes equations are symmetric. Have the authors tried using Conjugate

45   Gradient? I additionally worry that the Jacobi preconditioner is inadequate for problems with floating ice, e.g., Antarctica, as shown in the references by Tezaur et al. and additionally: (1) T. Isaac, G. Stadler, and O. Ghattas, Solution of nonlinear Stokes equations discretized by high-order finite elements on nonconforming and anisotropic meshes, with application to

ice sheet dynamics, SIAM J. Sci. Comput., 37 (2015), pp. B804–B833, doi:10.1137/140974407 and (2) R. Tuminaro, M. Perego, I. Tezaur, A. Salinger, and S. Price. A Matrix Dependent/Algebraic Multigrid Approach for Extruded Meshes with Applications to Ice Sheet Modeling. SIAM Journal on Scientific Computing 2016 38:5, C504- C532. Are more sophisticated preconditioners required for Antarctica? Is scaling of those comparable to what you show for Greenland?

Within our setup we have almost no floating ice (the SICOPOLIS simulation used as initial condition prevents the occurrence of floating ice; floating ice in our setup might occur due to the interpolation of SICOPOLIS data to the ISSM grid), so we are not suffering under these conditions. However, we have not observed a poor performance with GMRES+bjacobi once floating ice is included. We tested various solver+preconditioner combinations (BICGstab, ILU, ASM, etc), but GMRES+bjacobi shows equal results to other combinations.

– Line 131-132: the linear solver convergence tolerances stated here seem loose to me. Have the authors verified that the solutions they have obtained at all their mesh resolutions are sufficiently converged/accurate? Accuracy/verification is an important thing to establish prior to studying scalability/performance.

The linear iteration is stopped if two convergence criteria are satisfied, which we think is rather strict. We verified that this choice of tolerance values is sufficient using a number of standard benchmark problems and selected real-life setups.

– It is really great that you have set up a workflow and are doing performance monitoring! I agree that without this, it is inevitable that performance will be compromised in a big code with a lot of moving parts. It isn't entirely clear to me when the performance monitoring tests are run. Does it happen every time there is a PR merged into ISSM? Does it happen automatically or it must be run manually? Some further discussion of this is warranted. All I found was the following phrase: "it is quite feasible to periodically run an instrumented version of the code as part of the regular work of domain scientists", which suggests the performance testing is not run regularly or automatically, but perhaps I am misunderstanding.

Instrumentation-based profiling using Score-P can be easily applied by domain scientists and performance engineers. We have not implemented an automatic workflow which might run on every merged PR. Instead of this we provide a white list filter file for Score-P which can be used when performance monitoring is desired.

– I did not really find discussion of how the load-balancing is done of the mesh on which the simulation proceeds, and how the mesh gets updated in a time-dependent simulation modeling ice sheet evolution. How do you partition the mesh? Do you repartition every time the geometry (active mesh) changes? Or you partition a mesh including active and inactive cells once in the beginning? The latter approach has the potential of giving a lot of procs with no elements or poorly load-balanced meshes. Some discussion of this is warranted. I think the load imbalances you talk about in the paper have a different cause, unless I am misunderstanding.

ISSM distributes all elements evenly using ParMETIS on the MPI processes during the initialization. This does not take into account whether an element is currently active or inactive. Therefore, there is no load balancing or repartitioning. We

observed some processes which idle within the stress balance module, but other modules like the moving front module are load-balanced. First attempts to optimize the load balancing in the stress balance module lead to worse load balancing in other modules. Since the current load balancing does not have a strong impact on the scaling, we decided to not pursue this issue further.

85    – Is there any hope to improve the scalability of the matrix assembly?

There is some hope, but no easy fix:

– Currently, MatAssemblyBegin and MatAssemblyEnd are called right behind each other. However, it might be possible to perform some communication between these calls to exploit the asynchronous nature of these assemblies.

90    – We have seen that most of the time is spent in creating the initial matrix structure. Therefore, code performance would benefit if more matrices would be reused in a fashion similar to our modification of the stress balance vertical module.

– In the current implementation, the extrapolation in the moving front module is executed independently for each parameter. This step might run faster if all executions are combined in one large matrix or the matrix which is used for the first extrapolation can be reused for all following extrapolations.

95    – Finally, a hybrid (MPI + OpenMP) parallelization would reduce the number of MPI processes which are involved in the assembly, while still employing potentially a large number of threads.

– It would be interesting to compare ISSM performance to that of other open-source ice-sheet models based on the HO Stokes equations. I am not suggesting to do this in the paper, just commenting.

Yes, we fully agree and it would also be a good means to learn from each other and improve codes. Your comment is 100   very welcome!

Minor comments:

– Change "to solve" in the first line of the abstract to "solving".

done

– Line 12: remove commas around "thus".

105    done

– Line 25: I don't really understand the phrase, "standalone ice sheet projections are suffering from a large spread in climate forcing fields". I don't think "suffering" is the right word here. Please rephrase.

We will rephrase this to 'challenged by' in the revised version.

– After a colon, one does not use a capital letter. In many such instances, the colon should be replaced with a period.

110    done

– Lines 131-132: I suggest stating what is \epsilon_i here so the reader does not have to refer to the appendix.

Good idea! The revised version mentions the name of the covergence criterion in this sentence, too.

– Line 140: there is a space missing before "For profiling and tracing".

done

– Line 147: replace colon with period.

done

– Line 171: "data are stored" instead of "data is stored". Change "scalars, both" to "scalars. Both".

done

– Line 182: replace colon with period.

done

– Line 190: change "The instrumented" to "the instrumented".

done

– Line 193: change "Sampling" to "sampling".

done

– All strong scaling figures: please move the linear scaling line to be either below or above all the other curves. It's very hard to see it with all the lines on top of it (e.g., in Fig. 3).

This is a good point - in the revised version we will move the line so that it is clearly visible!

– Line 288: change "It" to "it".

done

– Line 293: change "12.000" to "12 000".

done

– Some of the figure captions say "(draft)". Was that intentional? I suspect it was not.

done

– Line 420: change "to very a modest" to "to a very modest".

done

**References**

Brædstrup, C. F., Damsgaard, A., and Egholm, D. L.: Ice-sheet modelling accelerated by graphics cards, Computers and Geosciences, 72, 210–220, https://doi.org/10.1016/j.cageo.2014.07.019, 2014.

---

## Referee Report (RR1)

**Review of revised article: A Scalability Study of the Ice-sheet and Sea-level System Model (ISSM, Version 4.18)**

**General Impression**

The authors have addressed most of the points I raised during the previous review. In particular, ambiguities concerning the relation of the here presented scalability study to preparations for exa-scale computing have been corrected. I still have a single remaining comment concerning the lines of arguments from the reply letter and one new paragraph on optimization levels. Else, **I see this article ready for publication**, suggesting this **last remaining point** to be addressed.

**Remaining points of critics**

As I wrote before, I think that the main points of criticism have been addressed. It is a valuable contribution to focus on pure MPI performance, but at the same time important to avoid any impression that this suffices to guarantee performance on modern CPU clusters. The only remaining concerning point is, that **for me you seem to motivate a pure MPI scalability study by rendering SIMD to be of limited importance**.

In an added paragraph and your reply letter you mention the insignificant performance gap between binaries with and without AVX512. Whereas I get the impression that you interpret this insignificant performance difference of a few percent as a justification to focus on MPI scalability, only, I cannot avoid the suspicion that this rather points to the situation that the investigated code does not sufficiently utilize the long vector units on modern CPU's, in particular during the matrix assembly.  In the end, one would need details on vector-unit memory utilization to draw a clear conclusion. In particular – referring to the justification in your letter of reply - I doubt that a pure MPI code without any SIMD optimization will excel on platforms like the A64FX CPUs in Fugaku (which derives the majority of its computing power from its 512-bit vectors). Most likely you will need the deployment of higher order elements (Isaac, et al., 2015) to effectively utilize those. With respect to modern architectures and even the memory-bounded nature of ISSM**,** I remain with my viewpoint that SIMD (AVX2/AVX512) performance is a to MPI scalability equally essential component to get a code performing on modern CPU clusters (e.g., Byckling, et al. 2017) and that it is important to not leave the reader with a possible interpretation that because of equal run-times of AVX512 enabled and disabled binaries SIMD would not be needed**.** Could you perhaps add a sentence on that aspect to clarify?

**Detailed list of remaining issues**

The list of issues is in the order of their occurrence in the text. Quote from the manuscript are kept in blue text.

**abstract** – **line 2**: In this paper, we discuss the scaling of the Ice-sheet and Sea-level System Model (ISSM) applied to the Greenland ice sheet with up to G250 resolution.

As the definition of the different abbreviation of runs is happening somewhere in the text (Table 1), in my opinion this should not be used in an abstract – my suggestion to make the abstract more informative on that point would be to simply write out the range of resolution used in G250.

**page 2 – line**: While MPI scalability is not sufficient for reaching exascale performance, it seems to be necessary as corroborated, for example, by the MPI-based programming style of the 24 applications chosen for the exascale computing project (see www.exascaleproject.org)
This is good information to add to the article. Yet, I tried to find the list of those 24 applications behind the link given here. I did not succeed. Could you please either specify the location with a more detailed link or – ideally, as links do not have DOI's – add a citable paper/report?

**page 6 – line 151:** The fact that we only use half of the available hardware cores is due to the fact that at G250 resolution, each MPI process requires 6.2 GB of memory.
I still think this would need the additional information that the excess amount of RAM is only needed in the lowest partition-count (=MPI task count) investigated – at least I presume this is the case. May I add that I still find it strange that you did not opt for a larger minimum partition amount or skipped the lowest partition number in order to completely fill all cores, which would resemble a more realistic production setup (and that is what you claim to investigate), as it is not economic to idle half of the resources one allocated for.

**page 6 – line 157:** We compiled both ISSM and PETSc with "-O2" as well as with "-O3 -march=cascadelake -mtune=cascadelake". On a 480-core configuration, the entire calculation (without loading the model), took 1955 seconds compiled with -O2, and 1930 seconds when compiled with higher-level optimization. With 1536 cores, we observed execution times of 763 and 748 seconds, respectively. So the compiler optimization level has an impact of less than 2% in both cases. As the impact of the more aggressive compiler optimizations is low, we stick to -O2 as it avoids some potential numerical issues that can arise with more aggressive compiler options.
**See my remaining point of critics above.** I would rather interpret this as a sign of underutilized vector units than to use it as an argument that it is of no impact.

**page 26 – line577:** For floating ice $h_b = -H\ \rho_i\rho_{sw}$ and $h_s = H(1-\ \rho_i\rho_{sw}$ with $\rho_{sw}$ being the sea water density.
Something went wrong with the typesetting of these two expressions.

**List of typos and to content less important issues**

**page 4 – line 117:** The balance equations for enthalpy and momentum are solved consecutively, no iteration between the two modules within a is done within a time step.
This sentence reads strange to me.

**page 4 – line 117:** There is a factor of about 34 between the minimum and the maximum DOF betweeen (-> between) G250 and G4000.
I also would swap the positions of "minimum" and "maximum" in the sentence, as maximum refers to G250 and minimum to G4000.

**page 5 – table 1:**
In the text (same page, line 140) you mention a minimum resolution of 150 m for G250, in the table you claim it to be from 0.25-10 km. A small inconsistency you might want to correct.
**page 22 – line 117:** we (-> We) also want to emphasize here, that we investigated the performance of an HO application only and that other issues may arise for other momentum balance choices.

**References**

Byckling, M., J. Kataja, M. Klemm, and T. Zwinger (2017): *OpenMP SIMD Vectorization and Threading of the Elmer Finite Element Software*, In: de Supinski B., Olivier S., Terboven C., Chapman B., Müller M. (eds) Scaling OpenMP for Exascale Performance and Portability. IWOMP 2017. Lecture Notes in Computer Science, vol 10468. Springer, doi: 10.1007/978-3-319-65578-9_9

Isaac, T., G. Stadler and O. Ghattas (2015): *Solution of Nonlinear Stokes Equations Discretized By High-Order Finite Elements on Nonconforming and Anisotropic Meshes, with Application to Ice Sheet Dynamics,* SIAM J. Sci. Comput., 37(6), B804–B833, doi: 10.1137/140974407

---

## Author Response (AR2)

**GMD Reviews and Authors' Response concerning the paper "A Scalability Study of the Ice-sheet and Sea-level System Model (ISSM, Version 4.18)"**

Yannic Fischler[1], Martin Rückamp[2], Christian Bischof[1], Vadym Aizinger[3], Mathieu Morlighem[4,5], and Angelika Humbert[2,6]

[1]Department of Computer Science, Technical University Darmstadt, Darmstadt, Hesse, Germany
[2]Alfred-Wegener-Institut, Helmholtz-Zentrum für Polar- und Meeresforschung, Bremerhaven, Bremen, Germany
[3]Chair of Scientific Computing, University of Bayreuth, Bayreuth, Bavaria, Germany
[4]Department of Earth Sciences, Dartmouth College, Hanover, United States of America
[5]Department of Earth System Science, University of California Irvine, United States of America
[6]Faculty of Geosciences, University of Bremen, Bremen, Germany

**Correspondence:** Yannic Fischler (yannic.fischler@tu-darmstadt.de)

*Copyright statement.* ©2021 all rights reserved

**1   General Comments**

We thank Thomas Zwinger for the second review. He highlighted some points, which we have clarified to not state out misleading information. In the following we answer each point of the review individually.

**2   Review by Thomas Zwinger**

**2.1   General Impression**

The authors have addressed most of the points I raised during the previous review. In particular, ambiguities concerning the relation of the here presented scalability study to preparations for exascale computing have been corrected. I still have a single remaining comment concerning the lines of arguments from the reply letter and one new paragraph on optimization levels. Else, **I see this article ready for publication**, suggesting this **last remaining point** to be addressed.

**2.2   Remaining points of critics**

As I wrote before, I think that the main points of criticism have been addressed. It is a valuable contribution to focus on pure MPI performance, but at the same time important to avoid any impression that this suffices to guarantee performance on modern CPU clusters. The only remaining concerning point is, that **for me you seem to motivate a pure MPI scalability study by**

15 **rendering SIMD to be of limited importance.** In an added paragraph and your reply letter you mention the insignificant performance gap between binaries with and without AVX512. Whereas I get the impression that you interpret this insignificant performance difference of a few percent as a justification to focus on MPI scalability, only, I cannot avoid the suspicion that this rather points to the situation that the investigated code does not sufficiently utilize the long vector units on modern CPU's, in particular during the matrix assembly. In the end, one would need details on vector-unit memory utilization to draw a clear

20 conclusion. In particular – referring to the justification in your letter of reply - I doubt that a pure MPI code without any SIMD optimization will excel on platforms like the A64FX CPUs in Fugaku (which derives the majority of its computing power from its 512-bit vectors). Most likely you will need the deployment of higher order elements (Isaac, et al., 2015) to effectively utilize those. With respect to modern architectures and even the memory-bounded nature of ISSM, I remain with my viewpoint that SIMD (AVX2/AVX512) performance is a to MPI scalability equally essential component to get a code performing on modern

25 CPU clusters (e.g., Byckling, et al. 2017) and that it is important to not leave the reader with a possible interpretation that because of equal run-times of AVX512 enabled and disabled binaries SIMD would not be needed. Could you perhaps add a sentence on that aspect to clarify?

**2.3 Detailed list of remaining issues**

The list of issues is in the order of their occurrence in the text. Quote from the manuscript are kept in blue text.

30 ### 2.3.1 abstract – line 2:

In this paper, we discuss the scaling of the Ice-sheet and Sea-level System Model (ISSM) applied to the Greenland ice sheet with up to G250 resolution.

As the definition of the different abbreviation of runs is happening somewhere in the text (Table 1), in my opinion this should not be used in an abstract – my suggestion to make the abstract more informative on that point would be to simply write out

35 the range of resolution used in G250.

You are right. We have rewritten the sentence to "... with horizontal grid resolutions varying between 10 and 0.25 km".

**2.3.2 page 2 – line:**

While MPI scalability is not sufficient for reaching exascale performance, it seems to be necessary as corroborated, for example, by the MPI-based programming style of the 24 applications chosen for the exascale computing project (see www.exascaleproject.org)

40 This is good information to add to the article. Yet, I tried to find the list of those 24 applications behind the link given here. I did not succeed. Could you please either specify the location with a more detailed link or – ideally, as links do not have DOI's – add a citable paper/report?

As we are not aware of any citable paper/report we specified the link: https://www.exascaleproject.org/about/

**2.3.3 page 6 – line 151:**

45  The fact that we only use half of the available hardware cores is due to the fact that at G250 resolution, each MPI process requires 6.2 GB of memory.

I still think this would need the additional information that the excess amount of RAM is only needed in the lowest partition-count (=MPI task count) investigated – at least I presume this is the case. May I add that I still find it strange that you did not opt for a larger minimum partition amount or skipped the lowest partition number in order to completely fill all cores, which 50 would resemble a more realistic production setup (and that is what you claim to investigate), as it is not economic to idle half of the resources one allocated for.

We have added additional information about the memory consumption.

Moreover, since ISSM is memory bound, the compute units of the CPUs are used more efficient while less threads/processes are running. Therefore the generalized statement "as it is not economic to idle half of the resources one allocated for" is not 55 tenable.

Since scaling differs in different scenarios (pure node scaling most likely shows different behavior than scaling on multiple nodes), we see single node execution with 48 processes as the base of our scaling analysis and did not want to exclude it. If we aimed to use all 96 processes per node, we would have to assume a minimum of about 480 processes.

**2.3.4 page 6 – line 157:**

60  We compiled both ISSM and PETSc with "-O2" as well as with "-O3 - march=cascadelake -mtune=cascadelake". On a 480-core configuration, the entire calculation (without loading the model), took 1955 seconds compiled with -O2, and 1930 seconds when compiled with higher-level optimization. With 1536 cores, we observed execution times of 763 and 748 seconds, respectively. So the compiler optimization level has an impact of less than 2% in both cases. As the impact of the more aggressive compiler optimizations is low, we stick to - O2 as it avoids some potential numerical issues that can arise with more aggressive 65 compiler options.

**See my remaining point of critics above.** I would rather interpret this as a sign of underutilized vector units than to use it as an argument that it is of no impact.

we agree that the impression should not be given that MPI scaling makes SIMD superfluous. Use of SIMD instructions is highly important for nodelevel performance which should be investigated separately. We added information to overcome the 70 missleading impression.

**2.3.5 page 26 – line 577:**

For floating ice $h_b = -H\rho_i\rho_{sw}$ and $h_s = H(1 - \rho_i\rho_{sw}$ with $\rho_{sw}$ being the sea water density.

Something went wrong with the typesetting of these two expressions.

Thanks. Typesetting of the equations is corrected.

**2.4 List of typos and to content less important issues**

**2.4.1 page 4 – line 118/119:**

The balance equations for enthalpy and momentum are solved consecutively, no iteration between the two modules within a is done within a time step.

This sentence reads strange to me.

We dropped the first occurrence of 'within a'.

**2.4.2 page 18 – line 364:**

There is a factor of about 34 between the minimum and the maximum DOF betweeen (-> between) G250 and G4000.

I also would swap the positions of "minimum" and "maximum" in the sentence, as maximum refers to G250 and minimum to G4000.

done

**2.4.3 page 5 – table 1:**

In the text (same page, line 140) you mention a minimum resolution of 150 m for G250, in the table you claim it to be from 0.25-10 km. A small inconsistency you might want to correct.

We have rewritten the sentence to: The highest resolution is ca. 250 m, which is close to the resolution of the bed topography dataset used in our study.

**2.4.4 page 22 – line 117:**

we (-> We) also want to emphasize here, that we investigated the performance of an HO application only and that other issues may arise for other momentum balance choices.

done

**2.5 References**

Byckling, M., J. Kataja, M. Klemm, and T. Zwinger (2017): OpenMP SIMD Vectorization and Threading of the Elmer Finite Element Software, In: de Supinski B., Olivier S., Terboven C., Chapman B., Müller M. (eds) Scaling OpenMP for Exascale Performance and Portability. IWOMP 2017. Lecture Notes in Computer Science, vol 10468. Springer, doi: 10.1007/978-3-319-65578-9_9

Isaac, T., G. Stadler and O. Ghattas (2015): Solution of Nonlinear Stokes Equations Discretized By High-Order Finite Elements on Nonconforming and Anisotropic Meshes, with Application to Ice Sheet Dynamics, SIAM J. Sci. Comput., 37(6), B804–B833, doi: 10.1137/140974407